# Effort drives saccade selection

**Damian Koevoet[1†], Laura Van Zantwijk[1], Marnix Naber[1], Sebastiaan Mathôt[2], Stefan van der Stigchel[1], Christoph Strauch[1*†]**

[1]Experimental Psychology, Helmholtz Institute, Utrecht University, Utrecht, Netherlands; [2]Department of Psychology, University of Groningen, Groningen, Netherlands

**\*For correspondence:**
c.strauch@uu.nl

[†]These authors contributed equally to this work

**Competing interest:** The authors declare that no competing interests exist.

## eLife Assessment

This study provides **important** findings on the nature of eye movement choices by human subjects. The study uses a novel approach and provides relatively clear and **convincing** results of the relationship between pupil size and saccade production. The results should be of interest to a broad audience interested in sensorimotor integration and sensory-guided decision-making.

**Abstract** What determines where to move the eyes? We recently showed that pupil size, a well-established marker of effort, also reflects the effort associated with making a saccade ('saccade costs'). Here, we demonstrate saccade costs to critically drive saccade selection: when choosing between any two saccade directions, the least costly direction was consistently preferred. Strikingly, this principle even held during search in natural scenes in two additional experiments. When increasing cognitive demand experimentally through an auditory counting task, participants made fewer saccades and especially cut costly directions. This suggests that the eye-movement system and other cognitive operations consume similar resources that are flexibly allocated among each other as cognitive demand changes. Together, we argue that eye-movement behavior is tuned to adaptively minimize saccade-inherent effort.

## Introduction

Humans make fast, ballistic eye movements, called saccades, to explore the rich visual world (*Findlay and Gilchrist, 2003*). Saccades are executed approximately three to four times per second (*Henderson, 2003*; *Henderson and Hollingworth, 1998*). Where to saccade is, therefore, one of the most frequent decisions the brain is faced with (*Bargary et al., 2017*).

It is well established that the physical properties of the environment (bottom-up information) (*Kümmerer et al., 2022*; *Kümmerer et al., 2016*; *Itti et al., 1998*; *Itti and Koch, 2001*; *Theeuwes, 1994*), the goals of the observer (top-down information) (*Posner, 1980*; *Petersen and Posner, 2012*; *Posner and Petersen, 1990*; *Desimone and Duncan, 1995*), and prior knowledge about a scene (selection history) (*Awh et al., 2012*; *Theeuwes et al., 2022*) drive where the eyes are moved. However, even when these factors are kept constant, there are many systematic biases in eye-movement behavior, such as a bias for cardinal compared with oblique saccade directions (*Foulsham and Kingstone, 2010*; *Gilchrist and Harvey, 2006*; *Bays and Husain, 2012*; *Anderson et al., 2008*; *Engbert and Kliegl, 2003*; *Tatler and Vincent, 2009*). The presence of these biases suggests that additional factors must contribute to the decision of where to saccade, here referred to as 'saccade selection.' Recent evidence suggests that the effort involved with planning and executing (eye) movements may be one crucial factor in driving action selection (*Shadmehr and Ahmed, 2020*; *Thomas et al., 2022*; *Hagura et al., 2017*; *Kadner et al., 2022*; *Cos et al., 2014*; *Cos et al., 2011*; *Cos et al., 2012*; *Todorov and Jordan, 2002*). Effort is thought to be minimized whenever possible (*Hull, 1943*;

*Tsai, 1932*), likely because it is costly to spend inherently limited cognitive resources (*Friston, 2010*; *Shadmehr and Ahmed, 2020*). We here use the term 'saccade cost' to describe the intrinsic effort associated with planning and executing saccades. Although saccades are relatively affordable [i.e. not very costly; *Findlay and Gilchrist, 2003*; *Theeuwes, 2012*], they are executed very often (*Henderson, 2003*; *Henderson and Hollingworth, 1998*) and, therefore, even small costs should add up over time (*Shadmehr and Ahmed, 2020*; *Koevoet et al., 2023*). We here hypothesized that affordable saccades are preferred over costly saccades. This would be in line with recent evidence from computational models suggesting that saccade costs predict saccade behavior [e.g. *Hoppe and Rothkopf, 2016*; *Hoppe and Rothkopf, 2019*; *Kadner et al., 2022*; *Thomas et al., 2022*]. These studies either assumed saccade costs or indirectly inferred them from gaze behavior itself. However, no study has been able to quantify saccade costs (neuro-)physiologically, and therefore been able to directly test this hypothesis until recently.

We recently demonstrated that the effort of saccade planning can be measured with pupil size, which allows for a physiological quantification of saccade costs as long as low-level visual factors are controlled for (*Koevoet et al., 2023*). Pupil size is an established marker of effort (*Kahneman, 1973*; *Bumke, 1911*; *Laeng et al., 2012*; *Mathôt, 2018*; *Strauch et al., 2022*; *Loewenfeld, 1993*; *Sirois and Brisson, 2014*; *van der Wel and van Steenbergen, 2018*; *Beatty, 1982*). For instance, loading more in working memory or tracking more objects results in stronger pupil dilation (*Beatty, 1982*; *Koevoet et al., 2024*; *Robison and Unsworth, 2019*; *Alnæs et al., 2014*; *Kahneman and Beatty, 1966*; *Unsworth and Miller, 2021*; *Unsworth and Robison, 2015*; *Ahern and Beatty, 1979*; *Hess and Polt, 1964*). Pupil size not only reflects cognitive (or mental) effort but also the effort of planning and executing movements (*Naber and Murphy, 2020*; *Richer and Beatty, 1985*; *Bumke, 1911*). We leveraged this to demonstrate that saccade costs can be captured with pupil size, and are higher for oblique compared with cardinal directions (*Koevoet et al., 2023*). Here, we addressed whether saccade costs predict where to saccade.

We hypothesized that participants would prefer affordable over costly saccades to minimize effort expenditure. To test this, we first mapped out saccade costs across directions by measuring pupil size during saccade planning. To assess saccade preferences across the same directions, a subsequent free choice saccade task was employed. Previewing our results, saccade costs indeed predicted saccade preferences, as affordable directions were preferred over costly alternatives. Strikingly, this general principle even held when participants searched for targets in natural scenes in two additional experiments: saccade cost remained a fundamental driver of saccade selection. If saccades and other cognitive operations consume the same resources, this should reflect in adaptively changing

saccade preferences in light of altering cognitive demands. We tested this idea experimentally by comparing saccade preferences with and without an auditory dual-task. As hypothesized, participants made fewer saccades overall under increased cognitive demand and especially cut the most costly directions. This provides convergent evidence that saccades are costly and rely at least in part on the same cognitive resources as other cognitively demanding operations.

## Results

### Saccade costs differ around the visual field

Twenty human participants planned and executed saccades in 36 different directions at a fixed amplitude (10°, *Figure 1a and b*). Pupil size was measured to index effort and thereby saccade cost during saccade planning (–150 ms until 170 ms around cue offset; also see *Appendix 1—figure 1*). Replicating our previous findings, we found that pupil size differed across directions (*Koevoet et al., 2023*; *Figure 1c–f*). We observed a larger pupil size during planning of oblique saccades compared with cardinal saccades (β = 7.662, SE = 1.957, $t$ = 3.916, p < 0.001). Downward saccades were associated with a larger pupil size than upward saccades (β = 0.556, SE = 0.171, $t$ = 3.261, p = 0.001), and we found a slightly larger pupil size for leftward compared with rightward saccades (β = 0.226, SE = 0.095, $t$ = 2.388, p = 0.017). These effects were not mediated by differences in saccade properties, such as duration, amplitude, peak velocity, and landing precision (*Figure 1e and f*). Together, this shows that saccade costs differ as a function of direction, indicating that certain saccades are more costly than others.

### Saccade costs predict saccade preferences

The same twenty participants subsequently completed a saccade preference task adapted from *Thomas et al., 2022*. To determine which of the 36 saccade directions were preferred, participants freely chose between two possible saccade targets in every trial (*Figure 2a*). We first analyzed whether saccade preferences differed across directions. Results showed that participants preferred cardinal over oblique directions (β = 0.091, SE = 0.023, $t$ = 3.910, p < 0.001; *Figure 2b*), and preferred upward over downward directions (β = 0.036, SE = 0.017, $t$ = 2.130, p = 0.033). No differences were observed between leftward and rightward saccade directions (β = 0.009, SE = 0.014, $t$ = 0.668, p = 0.504).

These results indicate that saccade preferences seem to mirror the pattern of saccade costs (compare *Figures 1d and 2b*). We proceeded to directly test if saccade costs predicted saccade preferences. To this end, we calculated the overall proportion that each direction was selected relative to how often it was offered to index saccade preferences (*Figure 2b*). In line with our hypothesis, pupil size during saccade planning (in cued directions) negatively correlated with saccade preferences (during self-selection) ($r$(34) = –0.76, p < 0.001; *Figure 2c*). For the first time, this demonstrates that intrinsic saccade costs critically predict saccade preferences. Put differently, participants preferred saccading towards affordable over costly options. A control analysis ruled out that the correlation between pupil size and saccade preferences was driven by other oculomotor metrics such as saccade latency and landing precision (see Supporting Information). Illustrating the robustness of this relationship further, we found smaller pupil sizes during saccade planning for preferred (selected >50%) than for avoided (selected <50%) directions ($t$(19) = 4.38, p < 0.001, Cohen's $d$ = 0.979; *Figure 2d*). As another test of the robustness of the effect, we analyzed whether saccade costs predicted saccade selection on a trial-by-trial basis. To this end, we first determined the more affordable option for each trial using the established saccade cost map (*Figure 1d*). We predicted that participants would select the more affordable option. Complementing the above analyses, the more affordable option was chosen above chance level across participants ($M$ = 56.64%, 95% CI = [52.75–60.52%], one-sample $t$-test against 50%: $t$(19) = 3.26, p = 0.004, Cohen's $d$ = 0.729; *Figure 2e*). Together, these analyses established that saccade costs robustly predict saccade preferences.

### Saccade costs predict saccade curvature and latency

If saccade cost is indeed weighed during saccade selection, this should be reflected in the oculomotor properties of the ensuing saccade. Saccade curvature reflects conflict between target and distractor saccade vectors: if a distractor is inhibited, and the target is activated, the saccade curves away from the distractor location (*Van der Stigchel et al., 2006*; *McPeek et al., 2003*; *Van der Stigchel, 2010*).

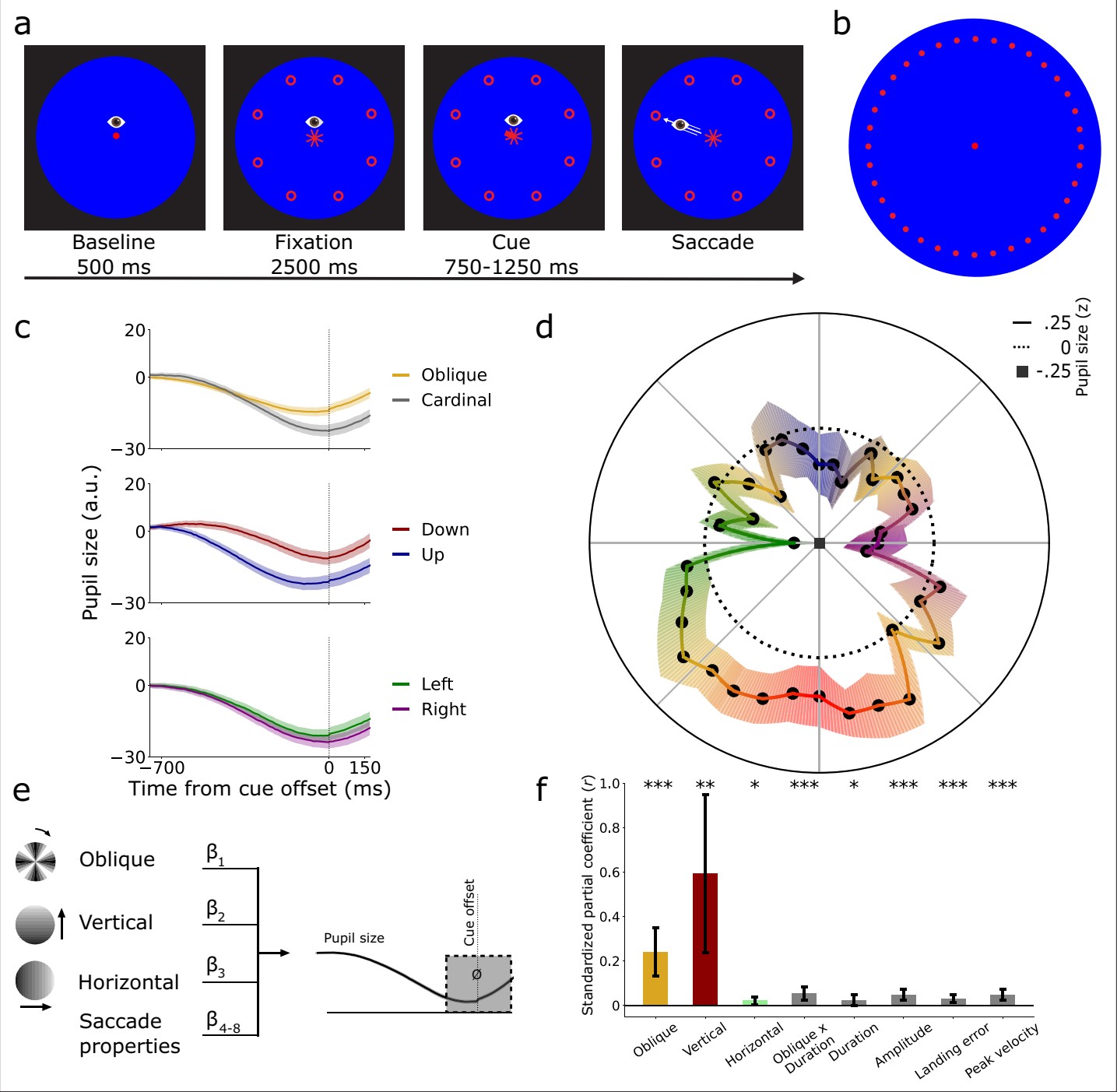

**Figure 1.** Pupil size differs during saccade planning across directions. (**a**) Twenty participants planned saccades in a cued direction. Saccades were executed as fast as possible upon cue offset. (**b**) All 36 possible saccade targets around the visual field. Only eight equally spaced locations were shown per trial. (**c**) Pupil size over time, split and averaged during saccade planning in oblique/cardinal, upward/downward, and left/rightward directions, locked to cue offset. Shaded areas indicate ± 1 s.e.m. (**d**) Averaged z-transformed pupil size during planning (–150 ms–170 ms post cue, gray area in **e**) across directions. (**e**) Linear mixed-effects model using obliqueness, verticalness, horizontalness of directions, and saccade properties to predict pupil size during saccade planning. (**f**) Standardized partial coefficients per predictor with 95% confidence intervals. *p < 0.05, **p < 0.01, ***p < 0.001.

Whenever saccade costs differ more between directions, there should, therefore, be more conflict between saccade vectors. If both directions were equally costly, there would be no need for conflict as cost minimization is impossible. We, therefore, hypothesized signs of increased oculomotor conflict to especially show in trials with relatively large differences in saccade costs. Furthermore, weighing costs

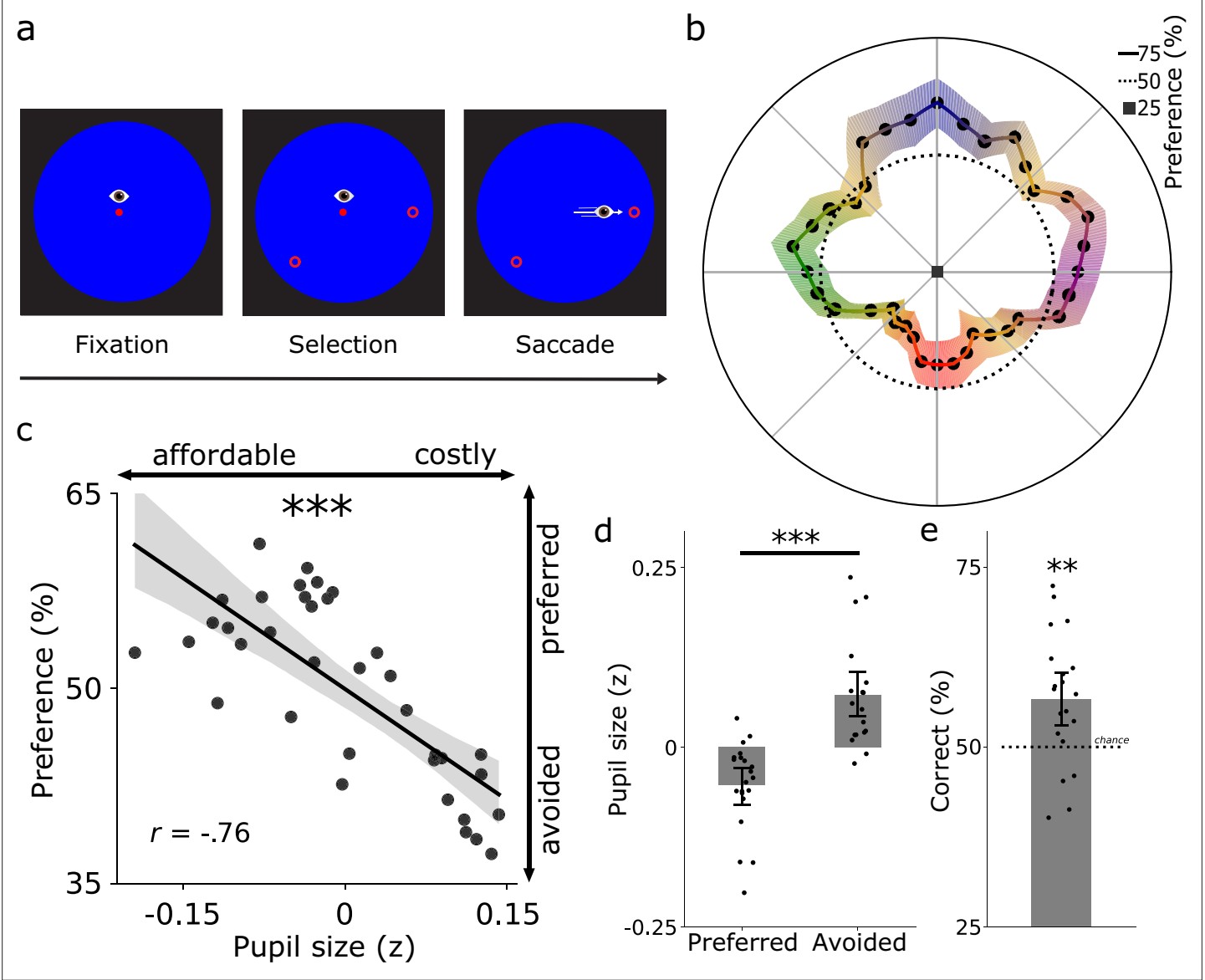

**Figure 2.** Saccade preferences differ across directions and are predicted by saccade costs. (**a**) The same twenty participants freely selected one of two saccade targets. (**b**) The average saccade preferences across directions (sum selected/sum offered). Shaded bands indicate ± 1 s.e.m. (**c**) Saccade costs correlated negatively with saccade preferences across directions: costly directions were avoided and affordable directions preferred. Black datapoints represent directions (averaged across participants). (**d**) Pupil size was larger for avoided compared with preferred directions. (**e**) Saccade costs predicted saccade selection on a trial-by-trial basis (56.64%). Together, the saccade costs in the first task predicted saccade preferences in the subsequent task. (**c-e**) Error bars reflect bootstrapped 95% confidence intervals. (**d-e**) Black datapoints represent participants. **p < 0.01, ***p < 0.001.

(and reward) in decision-making is known to take time (*Spering, 2022*, *Polanía et al., 2014*). More elaborate decisions should, therefore, not only show in more curvature, but also in longer saccade latencies.

To test this, we first split trials into saccades curving toward and away from the non-selected option. Saccades curved away from the non-selected option in the majority of trials, indicating oculomotor conflict (*Figure 3a*; M = 78.15%, 95% CI = [74.734–81.567%], t(19) = 15.741, p < 0.001, Cohen's d = 3.519). We then examined how saccade curvature and latency predicted the difference in pupil size between the two possible saccade targets. Whenever the difference in pupil size between the two options was larger, saccades curved away more from the non-selected option (β = 0.004, SE = 0.001, t = 4.448, p < 0.001; *Figure 3b*), and their latencies slowed (β = 0.050, SE = 0.013, t = 4.323, p < 0.001; *Figure 3c*). Our results show that cost is actively weighed and leads to stronger conflict whenever cost

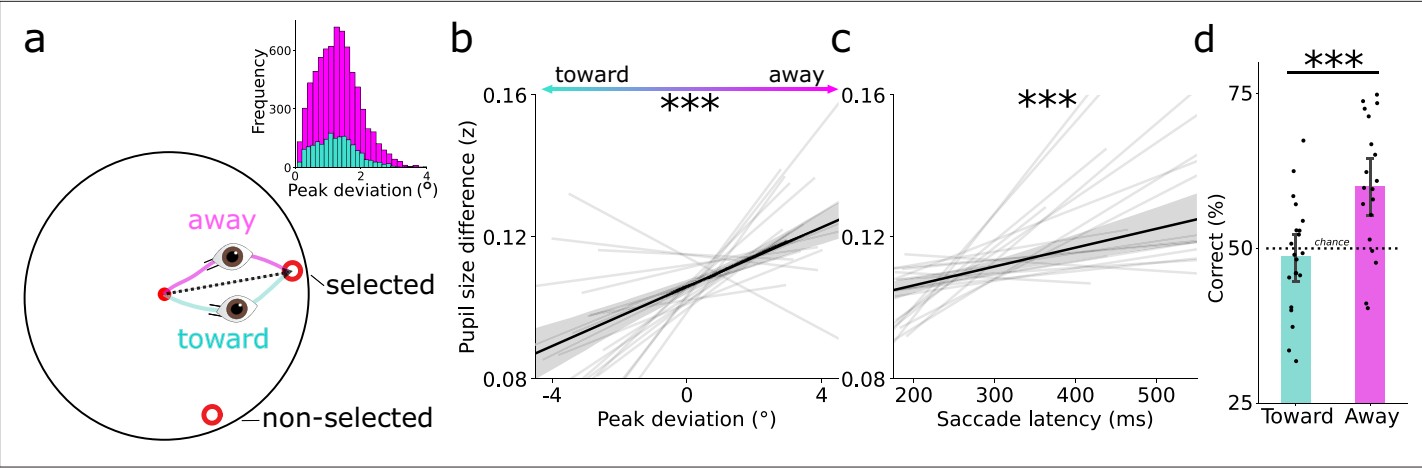

**Figure 3.** Saccade curvature and latency reveal active weighing of cost during saccade selection. (**a**) Schematic layout of saccade trajectories curving away (magenta) or toward (cyan) non-selected options. Curvature was calculated as the peak deviation from a straight line between gaze positions at saccade onset and offset. The top-right histogram shows that more saccades curved away than toward the non-selected option. (**b**) Difference in pupil size during the saccade planning task is linked to the peak curvature deviation in the saccade preference task. (**c**) Same as b, but now linked to saccade latency in the saccade preference task. Larger differences in pupil size are related to more oculomotor conflict between the two options, as reflected in more curvature away from the non-selected option and longer saccade latencies. (**b, c**) Black line depicts the relationship across all trials, gray lines denote regression fits per participant. (**d**) Saccade-cost based prediction of saccade selection split for toward and away curving trials. On a trial-by-trial basis, saccade costs predicted saccade selection above chance (59.72%) when saccades curved away from the non-selected option. In contrast, saccade costs did not predict saccade selection for 'toward' saccades. Black datapoints represent participants. All error bars reflect bootstrapped 95% confidence intervals. ***p < 0.001.

differences are larger. This suggests that more elaborate decisions in saccade selection are predominantly made when warranted by sufficient differences in saccade costs between options.

The above analyses show that saccade costs affect oculomotor conflict, but does increased conflict between saccade vectors also lead to selecting more affordable options? We expected that especially when oculomotor conflict was high, participants would choose the more affordable option. This means that saccade costs should be more predictive of saccade selection in away compared with toward curving trials. To test this idea, we repeated the trial-by-trial prediction of which option was selected as before (*Figure 2e*), but now separately for trials with toward and away curving saccades. Pupil size (i.e. saccade cost) predicted saccade selection when saccades curved away ($M = 59.72\%$, 95% CI = [55.208–64.233%], $t(19) = 4.116$, p < 0.001, Cohen's $d = 0.920$), but not toward ($M = 48.42\%$, 95% CI = [44.496–52.338%], $t(19) = 0.771$, $P = 0.450$, Cohen's $d = 0.172$) the non-selected option. These prediction accuracies differed between curve directions ($t(19) = 4.795$, p < 0.001, Cohen's $d = 1.072$; *Figure 3d*). This shows that saccade costs were predominantly considered when saccades curved away. Together, these analyses suggest that the costs of potential saccade targets are especially weighed during saccade selection when warranted by large differences in saccade costs. In these cases, oculomotor conflict increases and saccade cost plays a bigger role in saccade selection.

## Saccade costs predict saccade preferences in natural viewing

The previous results establish that saccade costs predict saccade preferences in highly controlled settings. However, a crucial question is whether saccade costs also predict saccade preferences in more complex and less controlled settings, in which physical saliency, the observer's goals, and prior knowledge about the scene also affect saccade selection. To test this, we analyzed data from two existing datasets (*Mathôt et al., 2015*) wherein participants (total n = 41) searched for small targets ('Z' or 'H') in natural scenes (*Figure 4a*; *Tkačik et al., 2011*). Again, we tested whether pupil size prior to saccades negatively linked with saccade preferences across directions. Because saccade costs and preferences across directions could differ for different situations (i.e. natural viewing vs. saccade preference task), but should always be negatively linked, we established both cost and preferences independently in each dataset. Many factors influence pupil size in such a natural task, for which we controlled as much as possible by including variables known to covary with pupil size in a linear

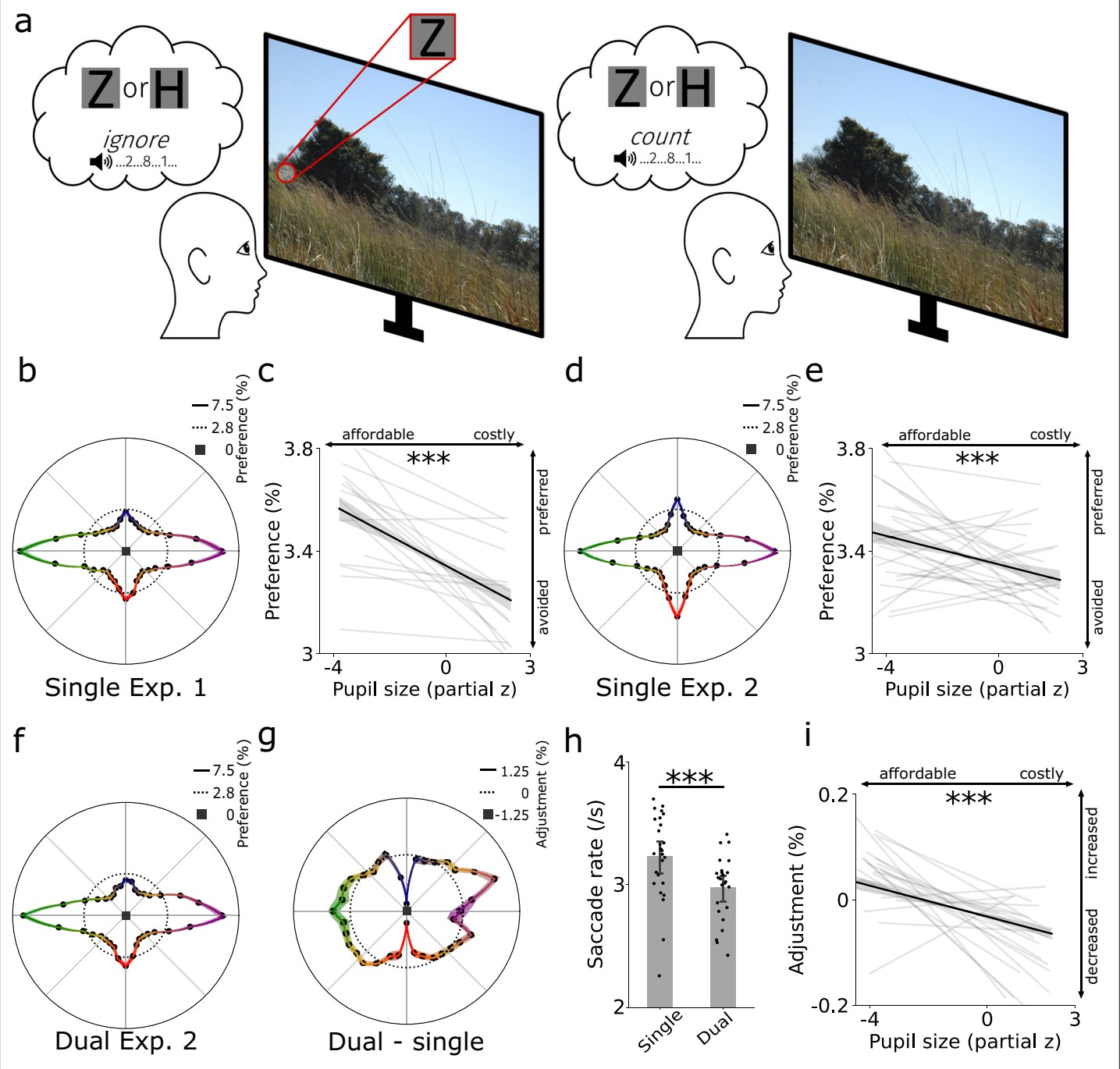

**Figure 4.** Saccade costs underlie saccade preferences in natural viewing. (**a**) Forty-one participants searched for small letters ('Z' or 'H') in natural scenes (Exp. 1; n = 16), and either ignored (single task) or additionally attended (dual task) an auditory number stream (Exp. 2; n = 25). (**b**) Saccade preferences during search without auditory stimulation. (**c**) Preferred directions were associated with a smaller pupil size prior to the saccade (Exp. 1). (**d, e**) Same as b, c but now for Exp. 2 without attending the auditory number stream (single task). Preferred directions were again associated with a smaller pupil size preceding the saccade (Exp. 2). (**f**) Same as d but now under the increased cognitive demand of the (primary) auditory digit counting (dual) task. (**g**) Adjustment in saccade preferences between single- and dual-task conditions in percentage points. (**h**) Less saccades were executed in the more demanding dual-task condition. Black datapoints represent participants. (**i**) Pupil size during the single task predicted direction adjustments under additional cognitive demand. Costly saccades as assessed during the single-task condition were especially cut in the dual-task condition. (**b, d, f, g**) Shaded bands represent ± 1 s.e.m. Other error bars reflect bootstrapped 95% confidence intervals. (**c, e, i**) Black lines depict the relationship across all trials, gray lines denote regression fits per participant. ***p < 0.001.

mixed-effects model (based on *Mathôt et al., 2015*; e.g. luminance, gaze position, saccade properties, saliency, fixation number; see Methods) to access the underlying saccade costs. As hypothesized, we observed a negative relationship between pupil size and saccade preferences in both experiments (Exp. 1: β = 1.784, SE = 0.324, $t$ = 5.412, p < 0.001; saccade preferences in *Figure 4b*, link in *Figure 4c*; Exp. 2: β = 0.644, SE = 0.170, $t$ = 3.780, p < 0.001; saccade preferences in *Figure 4d*, link in *Figure 4e*). This shows that even when participants made unconstrained eye movements in natural scenes, saccade cost remained linked to saccade preferences: affordable directions were preferred over costly directions.

Do cognitive operations and eye movements consume from a similar pool of resources (*Beatty, 1982*)? If so, increasing cognitive demand for non-oculomotor processes should result in decreasing available resources for the oculomotor system. In line with this idea, previous work indeed shows altered eye-movement behavior under effort as induced by dual tasks, for example by making less saccades under increased cognitive demand (*May et al., 1990*; *Walter and Bex, 2021*; *Cui and Herrmann, 2023*). We, therefore, investigated whether less saccades were made as soon as participants had to count the occurrence of a specific digit in the auditory number stream in comparison to ignoring the stream (in Exp. 2; *Figure 4a*). Participants were instructed to prioritize the auditory digit-counting task over finding the visual search target. Therefore, resources should be shifted from the oculomotor system to the primary auditory counting task. The additional cognitive demand of the dual task indeed led to a decreased saccade frequency ($t$(24) = 7.224, p < 0.001, Cohen's $d$ = 1.445; *Figure 4h*). This indicates that the auditory dual task and the oculomotor system, at least in part, consumed from a shared pool of cognitive resources.

From a costs-perspective, it should be efficient to not only adjust the number of saccades (non-specific), but to also cut especially expensive directions the most (specific). Therefore, we expected participants to especially avoid costly saccades (as assessed in the single task) under higher cognitive demand (induced by the dual task). We calculated a saccade-adjustment map (*Figure 4g*) by subtracting the saccade preference map in the single task (*Figure 4f*) from the dual-task map (*Figure 4d*). Participants seemingly cut vertical saccades in particular, and made more saccades to the top right direction. This pattern may have emerged as vertical saccades are more costly than horizontal saccades (also see *Figure 1d*). Oblique saccades may not have been cut because there were very little oblique saccades in the single condition to begin with (*Figure 4d*), making it difficult to observe a further reduction of such saccades under additional cognitive demand (i.e. a floor effect). Nevertheless, pupil size negatively linked with the adjustment map as hypothesized (β = 9.333, SE = 0.966, $t$ = 9.659, p < 0.001; *Figure 4i*; while controlling for the same possible covariates as before). This shows that costly saccades were cut disproportionally when more cognitive resources were consumed by the additional auditory dual task. This demonstrates that cognitive resources are flexibly (dis)allocated from and to the oculomotor system based on the current resource demands.

## Discussion

We here investigated whether effort determines saccade preferences. We first measured pupil size prior to saccade execution across directions as a physiological marker of effort and thus saccade costs. Next, saccade preferences were assessed in the same participants and directions. We observed that affordable saccades were preferred over costly ones. This is especially remarkable given that the delayed saccades in the planning task likely differ in their oculomotor program from the immediate saccades in the preference task in some regard. Furthermore, when two possible saccade directions differed more in saccade cost, we found higher oculomotor conflict as indexed by stronger saccade trajectory deviations away from the non-selected option and increased onset latencies. In two additional experiments, we demonstrated the link between saccade costs and saccade preferences to be robust even when participants made unconstrained eye movements during natural viewing. Lastly, saccade directions were flexibly adjusted based on cost as cognitive demand increased. Together, this demonstrates that saccade costs fundamentally underlie saccade selection, even when physical salience, the goals of the observer, and selection history affect where the eyes are moved.

What contributes to intrinsic saccade costs? We speculate that at least three processes contribute to the total cost of a saccade (*Koevoet et al., 2023*): the complexity of oculomotor programming (*Jainta et al., 2011*; *Wang and Munoz, 2021*; *Shadmehr and Ahmed, 2020*), shifting of presaccadic attention (*Rizzolatti et al., 1987*; *Deubel and Schneider, 1996*), and predictive/spatial remapping

(*Rolfs, 2015*; *Melcher, 2007*; *Bays and Husain, 2007*; *Fabius et al., 2019*). The complexity of an oculomotor program is arguably shaped by its neural underpinnings. For example, oblique but not cardinal saccades require communication between pontine and midbrain circuits (*Sparks, 2002*; *King and Fuchs, 1979*; *Curthoys et al., 1984*). Such differences in neural complexity may underlie the additional costs of oblique compared with cardinal saccades. Besides saccade direction, other properties of the ensuing saccade such as its speed, distance, curvature, and accuracy may contribute to a saccade's total cost (*Shadmehr and Ahmed, 2020*; *Naber and Murphy, 2020*; *Wang et al., 2017*; *Koevoet et al., 2023*; *Wang et al., 2021*) but this remains to be investigated directly. Furthermore, presaccadic attention is shifted prior to each saccade to prepare the brain for the abrupt changes in retinal input resulting from saccades through spatial/predictive remapping (*Melcher and Colby, 2008*; *Rolfs, 2015*; *Melcher, 2007*; *Bays and Husain, 2007*; *Rizzolatti et al., 1987*; *Deubel and Schneider, 1996*; *Van der Stigchel and Hollingworth, 2018*). This preparation for upcoming changes in retinal input consumes neurocognitive resources and, therefore, likely also contributes to saccade costs (*Koevoet et al., 2023*; *Shadmehr and Ahmed, 2020*). To better understand saccade selection more generally, future work should elucidate which processes contribute to saccade costs, and how costs shape different (aspects of) saccades.

The observed differences in saccade costs across directions could be linked to established anisotropies in perception (*Wexler et al., 2022*; *Barbot et al., 2021*; *Baldwin et al., 2012*; *Greenwood et al., 2017*; *Chakravarthi et al., 2022*; *Abrams et al., 2012*; *Ezzo et al., 2023*), attention (*Hanning et al., 2024*; *Hanning et al., 2022*; *Schütz, 2014*; *Carrasco et al., 2001*; *Mackeben, 1999*; *Ohl et al., 2024*), saccade characteristics (*Hanning et al., 2024*; *Greenwood et al., 2017*; *Hanning et al., 2022*; *Ohl et al., 2024*), and (early) visual cortex (*Himmelberg et al., 2022*; *Van Essen et al., 1984*; *Benson et al., 2021*; *Himmelberg et al., 2021*; *Silva et al., 2018*; also see *Himmelberg et al., 2023*). For example, downward saccades are more costly than upward saccades, which mimics a similar asymmetry in early visual areas wherein the upper visual field is relatively underrepresented (*Himmelberg et al., 2022*; *Van Essen et al., 1984*; *Benson et al., 2021*; *Himmelberg et al., 2021*; *Silva et al., 2018*); similarly stronger presaccadic benefits are found for down- compared with upward saccades (*Hanning et al., 2024*, *Hanning et al., 2022*). Moreover, upward saccades are more precise than downward saccades (*Greenwood et al., 2017*). Future work should elucidate where saccade cost or the aforementioned anisotropies originate from and how they are related - something that pupil size alone cannot address.

We here measured cost as the degree of effort-linked pupil dilation. In addition to pupil size, other markers may also indicate saccade costs. For example, saccade latency has been proposed to index oculomotor effort (*Shadmehr et al., 2019*), whereby saccades with longer latencies are associated with more oculomotor effort. This makes saccade latency a possible complementary marker of saccade costs (also see Appendix). Although relatively sluggish, pupil size is a valuable measure of attentional costs for (at least) two reasons. First, pupil size is a highly established marker of effort, and is sensitive to effort more broadly than only in the context of saccades (*Sirois and Brisson, 2014*; *Strauch et al., 2022*; *Kahneman, 1973*; *Kahneman and Beatty, 1966*; *Beatty, 1982*; *Laeng et al., 2012*; *Bumke, 1911*; *van der Wel and van Steenbergen, 2018*; *Koevoet et al., 2024*; *Loewenfeld, 1993*; *Mathôt, 2018*). Pupil size, therefore, allows to capture not only the costs of saccades, but also of covert attentional shifts (*Koevoet et al., 2023*), or shifts with other effectors such as head or arm movements (*Richer and Beatty, 1985*; *Voudouris et al., 2023*). Second, as we have demonstrated, pupil size can measure saccade costs even when searching in natural scenes (*Figure 4*). During natural viewing, it is difficult to disentangle fixation duration from saccade latencies, complicating the use of saccade latency as a measure of saccade cost. Together, pupil size, saccade latency, and potential other markers of saccade cost could fulfill complementary roles in studying the role of cost in saccade selection.

Our findings are in line with established effort-based models that assume costs to be weighed against rewards during decision-making (*Kool et al., 2010*; *Kurzban et al., 2013*; *Shenhav et al., 2021*; *Shenhav et al., 2013*; *Shenhav et al., 2017*; *Westbrook and Braver, 2015*). In such studies, reward and cognitive/physical effort are often parametrically manipulated to assess how much effort participants are willing to exert to acquire a given (monetary) reward (e.g. *Bustamante et al., 2023*, *Müller et al., 2022*). Whereas this line of work manipulated the extrinsic costs and/or rewards of decision options (e.g. perceptual consequences of saccades [*Lisi et al., 2019*; *Sedaghat-Nejad and*

*Shadmehr, 2021*] or consequences associated with decision options), we here focus on the intrinsic costs of the movement itself (in terms of cognitive and physical effort). Relatedly, the intrinsic costs of arm movements are also considered during decision-making: biomechanically affordable movements are generally preferred over more costly ones (*Cos et al., 2011*; *Cos et al., 2014*; *Cos et al., 2012*). We here extend these findings in two important ways. First, until now, the intrinsic costs of saccades and other movements have been inferred from gaze behavior itself or by using computational modeling (*Thomas et al., 2022*; *Kadner et al., 2022*; *Hoppe and Rothkopf, 2016*; *Hoppe and Rothkopf, 2019*; *Cos et al., 2011*; *Cos et al., 2014*; *Cos et al., 2012*; *Petitet et al., 2021*). In contrast, we directly measured cost physiologically using pupil size. Second, we show that physiologically measured saccade costs predict where saccades are directed in a controlled binary preference task, and even during natural viewing. Our findings could unite state-of-the-art computational models [e.g. *Kadner et al., 2022*; *Thomas et al., 2022*; *Hoppe and Rothkopf, 2016*; *Hoppe and Rothkopf, 2019*; *Tatler et al., 2017*] with physiological data, to directly test the role of saccade costs and ultimately further our understanding of saccade selection.

Throughout this paper, we have used cost in the limited context of saccades. However, cost-based decision-making may be a more general property of the brain (*Friston, 2010*; *Kahneman, 1973*; *Attneave, 1959*; *Hasenstaub et al., 2010*; *Jamadar et al., 2024*). Every action, be it physical or cognitive, is associated with an intrinsic cost, and pupil size is likely a general marker of this (*Beatty, 1982*). Note, however, that pupil dilation does not always reflect cost, as the pupil dilates in response to many sensory and cognitive factors which should be controlled for, or at least considered, when interpreting pupillometric data [e.g., see *Mathôt and Vilotijević, 2023*; *Strauch et al., 2022*; *Mathôt, 2018*; *Sirois and Brisson, 2014*]. Effort-linked pupil dilations are thought to be, at least in part, driven by activity in the brainstem locus coeruleus (LC) (*Joshi et al., 2016*; *Joshi and Gold, 2020*; *Aston-Jones and Cohen, 2005b*; *Strauch et al., 2022*) [but other neurotransmitters also affect pupil size, e.g. *Grujic et al., 2024*, *Mazancieux et al., 2023*]. Activity in LC with its widespread connections throughout the brain (*Schwarz and Luo, 2015*; *Szabadi, 2013*; *Aston-Jones and Cohen, 2005b*; *Berridge and Waterhouse, 2003*; *Aston-Jones and Waterhouse, 2016*; *Aston-Jones and Cohen, 2005a*) is considered to be crucial for the communication within and between neural populations and modulates global neural gain (*Poe et al., 2020*; *Wainstein et al., 2022*; *Dahl et al., 2022*; *Corbetta et al., 2008*; *Posner et al., 2006*). Neural firing is costly (*Shadmehr and Ahmed, 2020*; *Attwell and Laughlin, 2001*), and therefore LC activity and pupil size are (neuro)physiologically plausible markers of cost (*Strauch et al., 2022*). Tentative evidence even suggests that continued exertion of effort (accompanied by altered pupil dilation) is linked to the accumulation of glutamate in the lateral prefrontal cortex (*Wiehler et al., 2022*), which may be a metabolic marker of cost [also see *Castrillon et al., 2023*; *Wiehler et al., 2022*; *Jamadar et al., 2024*].

Besides the costs of increased neural activity when exerting more effort, effort should be considered costly for a second reason: Cognitive resources are limited. Therefore, any unnecessary resource expenditure reduces cognitive and behavioral flexibility (*Friston, 2010*; *Shadmehr and Ahmed, 2020*; *Kahneman, 1973*; *Jamadar et al., 2024*). As a result, the brain needs to distribute resources between cognitive operations and the oculomotor system. We found evidence for the idea that such resource distribution is adaptive to the general level of cognitive demand and available resources: Increasing cognitive demand through an additional primary auditory dual task led to a lower saccade frequency, and especially costly saccades were cut. In this case, it is important to consider that the auditory task was the *primary task*, which should cause participants to distribute resources from the oculomotor system to the counting task. In other situations, more resources could be distributed to the oculomotor system instead, for example, to discover new sources of reward (*Shadmehr and Ahmed, 2020*; *Gottlieb, 2012*). Adaptive resource allocation from, and to the oculomotor system parsimoniously explains a number of empirical observations. For example, higher cognitive demand is accompanied by smooth pursuits deviating more from to-be tracked targets (*Kosch et al., 2018*), reduced (micro)saccade frequencies (*Figure 4*; *Siegenthaler et al., 2014*; *May et al., 1990*; *Walter and Bex, 2021*; *Cui and Herrmann, 2023*), and slower peak saccade velocities (*Di Stasi et al., 2011*; *App and Debus, 1998*; *Sylvestre and Cullen, 1999*). Relatedly, more precise saccades are accompanied with worse performance in a crowding task (*Greenwood et al., 2017*). Furthermore, it has been proposed that saccade costs are weighed against other cognitive operations such as using working memory (*Somai et al., 2020*; *Van der Stigchel, 2020*; *Hoogerbrugge et al., 2023*; *Koevoet et al.,*

2023; *Ballard et al., 1995*). How would the resources between the oculomotor system and cognitive tasks (like the auditory counting task) be related? One possibility is that both consume from limited working memory resources (*Luck and Vogel, 2013*; *Ma et al., 2014*). Saccades are thought to encode target objects in a mandatory fashion into (visual) working memory (*Van der Stigchel and Hollingworth, 2018*), and the counting task requires participants to keep track of the auditory stream and maintain the count of the instructed digit in working memory. However, the exact nature of which resources overlap between tasks remain open for future investigation (also see *Nozari and Martin, 2024*). Together, we propose that cognitive resources are flexibly (dis)allocated to and from the oculomotor system based on the current demands to establish an optimal balance between performance and cost minimization.

How does the brain keep track of saccade costs, and which areas use it during saccade selection? Although our data do not allow direct inferences about the precise neural circuitry underlying the computations of oculomotor selection, oculomotor control is generally thought to be steered by a network encompassing the frontal eye field (*Paus, 1996*; *Bruce et al., 1985*), the supplementary eye field (*Schlag and Schlag-Rey, 1987*; *Amiez and Petrides, 2009*; *Sharika et al., 2013*), the anterior cingulate cortex (*Gaymard et al., 1998*; *Ruehl et al., 2021*; *Pouget, 2015*; *Conti and Irish, 2021*), the superior colliculus (*Gandhi and Katnani, 2011*; *Sparks, 1999*; *Sparks, 2002*; *Glimcher and Sparks, 1992*; *Wurtz and Albano, 1980*; *Basso and May, 2017*; *Strauch et al., 2022*), and the cerebellum (*Voogd and Barmack, 2006*; *Tanaka et al., 2021*; *Takagi et al., 1998*; *Van der Stigchel et al., 2006*). These areas are not just associated with oculomotor control, but are all also thought to be crucial for decision-making processes (*Zhang et al., 2021*; *Wang et al., 2020*; *Basso and May, 2017*; *Jun et al., 2021*; *Crapse et al., 2018*; *Deverett et al., 2018*; *Basso et al., 2021*; *So and Stuphorn, 2016*; *Yang and Heinen, 2014*; *Chudasama et al., 2013*; *Baumann et al., 2015*; *Schall, 2001*). It is plausible that the weighing of saccade costs during saccade selection is performed by this oculomotor-decision making network, but other areas such as orbitofrontal cortex may also play a role (*Padoa-Schioppa and Conen, 2017*; *Wallis, 2007*).

We report a combination of correlational and causal findings. Despite the correlational nature of some of our results, they consistently support the hypothesis that saccade costs predict saccade selection [which we predicted previously, *Koevoet et al., 2023*]. Causal evidence was provided by the dual-task experiment as saccade frequencies - and especially costly saccades were reduced under additional cognitive demand. Only a cost account predicts (1) a link between pupil size and saccade preferences, (2) a cardinal saccade bias, (3) reduced saccade frequency under additional cognitive demand, and (4) disproportional cutting of especially those directions associated with more pupil dilation. Together, our findings converge upon the conclusion that effort drives saccade selection.

To conclude, we have demonstrated that saccade costs can be measured using pupil size and that these costs robustly predict saccade selection. We propose that saccade selection is driven by physical properties of the environment, the observer's goals, selection history, and another fundamental factor: effort.

## Methods
### Saccade planning and saccade preference tasks
#### Participants
Twenty-two participants with normal or corrected-to-normal vision took part in the saccade planning and preference tasks across two sessions. One participant was excluded due to only finishing a single session, and another dataset was discarded due to not following task instructions (<50% included trials in the saccade planning task). Twenty participants were included in the analyses for the saccade planning, and saccade preference tasks (age: $M = 24.00$, range: (19-31), 12 women, 8 men). The current sample size was comparable with previous work investigating saccade costs (*Koevoet et al., 2023*; *Thomas et al., 2022*). The total number of trials was substantially larger in the current dataset than in *Koevoet et al., 2023* (14,400 vs 4800), albeit from a slightly smaller number of participants (n = 20 vs. n = 24). Participants provided written informed consent before taking part, and were awarded monetary compensation or course credits. The experimental procedure was approved by Utrecht University's ethical review board of the Faculty of Social Sciences (22–0635).

## Apparatus and stimuli

Gaze position and pupil size were recorded at 1000 Hz with an Eyelink 1000 desktop mount (SR Research, Ontario, Canada) in a brightness- and sound-attenuated laboratory. A chin- and forehead-rest limited head movements. Stimuli were presented using PsychoPy [v.2022.2.5; *Peirce et al., 2019*] on an ASUS ROG PG278Q monitor (2560 × 1440, 100 Hz) positioned 67.5 cm away from eye position. The eye-tracker was calibrated (9 points) at the beginning of each session, during each break, and whenever necessary throughout the experiment (same procedure for both tasks, see below).

Potential saccade targets were eight equally spaced out red rings (1° diameter) positioned at an eccentricity of 10° visual angle. The central fixation stimulus was a red eight-legged asterisk of which each leg pointed towards one of the possible saccade targets (1°). These stimuli were presented on a blue circle (12° diameter; 11.64 cd/m$^2$); the remaining part of the screen was black (0.73 cd/m$^2$) to ensure equal brightness across all 36 possible target locations (*Figure 1a and b*).

To control for low-level visual effects on pupil size, the red color of all stimuli was made equilumi-nant to the blue background color using a flicker fusion calibration [as in *Koevoet et al., 2023*]. A blue background (HSV: 240.1.1) was presented continuously while a central red circle (5° diameter) contin-uously flickered at 25 Hz. Participants adjusted the luminance of the red color by moving the mouse across the horizontal plane of the screen until the flickering was the least noticeable, and then clicked the left mouse button to confirm. This procedure was performed thrice, and the average luminance of the red color was used for the fixation and target stimuli throughout the task. Participants completed the flicker fusion calibration preceding each task.

## Procedure

The experiment started with the saccade planning task (*Koevoet et al., 2023*), wherein participants planned saccades into 36 different cued directions. Each trial started when the central stimulus was fixated for 500 ms. After a fixation period (2000 ms) eight equally spaced potential saccade targets were presented (randomized which eight out of the 36 between trials). Afterwards, one of the eight legs of the asterisk became slightly thicker, cueing a saccade target (750–1250 ms, 100% valid). Partic-ipants planned and withheld an eye movement until cue offset, and then executed the saccade as fast as possible. Trials ended upon fixating the target stimulus (within 3°) for 500 ms. Whenever partic-ipants saccaded too early or to an incorrect location, red feedback text was presented ('too early,' 'wrong location'). Each session consisted of 360 trials, preceded by ten practice trials. Participants could initiate a break whenever they wanted.

Participants subsequently completed a saccade preference task [adapted from *Thomas et al., 2022*]. Upon briefly fixating the central stimulus (10–500 ms), possible saccade targets were presented in two out of the 36 positions around the visual field. The only restriction was that the two targets should at least differ 20° in angle to ensure targets were sufficiently spaced out - and to limit saccade averaging (*Van der Stigchel and Nijboer, 2013*). Trials ended when one of the two saccade targets was fixated for 50ms. Participants completed 360 trials per session.

## Data processing and analyses

All data were analyzed using custom Python (v3.9.14) and R (v4.3.1) scripts. Analyses of pupillometric data followed recommendations by *Strauch et al., 2022*; *Mathôt and Vilotijević, 2023*. Blinks were interpolated (*Mathôt and Vilotijević, 2023*), data were downsampled to 100 Hz, and pupil data were subtractively baseline corrected with the mean of the first 250 ms after cue onset. Saccades were detected offline using an onset velocity threshold of 75 °/s and an offset threshold of 1 °/s. Trials with fast (<175 ms) or slow (>550 ms) onset latency (*Koevoet et al., 2023*), a very short (<10 ms) or long saccade (>110 ms) duration (*Nyström and Holmqvist, 2010*), an amplitude smaller than 5°, saccades landing more than 2° from the target location, saccades toward the wrong target, and practice trials were discarded (13.67% in total). To map out saccade planning costs across directions, the average pupil size was calculated 150 ms before until 170 ms after cue offset (*Figure 1f*) - before any saccade onsets to prevent pupil foreshortening errors (*Hayes and Petrov, 2016*). We analyzed saccade costs by incorporating continuous predictors for oblique (cardinal vs. oblique; 0–4), vertical (up vs. down; in y coordinates), and horizontal (left vs. right; in x coordinates) direction biases in a linear mixed-effects model (*Figure 1e and f*). We also incorporated properties of the ensuing saccade to control for their possible associations with pupil size. The final model was determined using AIC-based backward

model selection (Wilkinson notation: pupil size ~ oblique*saccade duration + vertical + horizontal + amplitude + landing error + peak velocity + (1 + oblique + vertical|participant)). For all mixed-effects models, we included as many by-participant random slopes as possible for our main effects of interest while ensuring model convergence (*Barr, 2013*; *Mathôt et al., 2015*). Absolute effect sizes (i.e. *r*) and their corresponding 95% confidence intervals for the linear mixed-effects models were calculated using *t* and *df* values with the 'effectsize' package (v.0.8.8) in R. To obtain an average saccade costs map, pupil sizes were z-transformed per participant within sessions, and then averaged across participants for each direction (*Figure 1d*).

For the saccade preference task, saccades were detected as above. Which saccade target was selected per trial was determined using the last 50 ms of gaze data of each trial - the option closest to the gaze position was treated as the selected target. Trials were discarded if the difference in distance between the two saccade options in gaze position was less than 1.5° (3.47%). A logistic mixed-effects model was fit to investigate anisotropies across directions (Wilkinson notation: saccade preference ~ oblique + vertical + horizontal + (1+oblique + vertical + horizontal|participant)). Saccade preference for each direction was calculated per participant by summing how often a direction was chosen and then dividing by the number of times that direction was offered. As for the saccade cost map, the average saccade preference map was obtained by averaging across participants (*Figure 2b*). Preferred (>50%) and avoided (<50% chosen) directions were grouped using the average preference map (*Figure 2d*). To investigate if cost predicted saccade selection on a trial-by-trial basis, we compared the saccade costs of the two potential options. We predicted that the option with the smaller pupil size from the average cost map (obtained from the saccade planning task) would be chosen. This procedure was performed for each participant, and subsequently tested against chance performance (50%) with a one-sample *t*-test.

Saccade curvature was computed using the peak deviation from a straight line between gaze position at saccade onset until saccade offset (*Figure 3a*; *Van der Stigchel et al., 2006*). Trials were excluded from these analyses if: (a) saccade latencies were shorter than 175 ms, (b) saccade amplitudes were smaller than 5°, (c) saccade durations were shorter than 10 ms or longer than 110 ms and (d) if the angle between targets exceeded 150° (*Nyström and Holmqvist, 2010*; *Koevoet et al., 2023*; *Van der Stigchel and Nijboer, 2013*). To analyze the relationship between saccade costs and saccade properties, we first computed the absolute saccade cost difference for each trial (as indexed from the average saccade cost map). A linear mixed-effects model was conducted to test whether saccade curvature and latency linked to saccade costs (Wilkinson notation: cost difference ~ peak deviation + latency + (1 + peak deviation|participant)). We then split the data based on whether saccades curved toward or away from the non-selected option. The same trial-by-trial analysis as described above was used to investigate if cost predicted saccade selection in toward and away trials separately.

## Search in natural scenes

### Procedure
We analyzed existing data of two experiments to investigate if effort drives saccade selection in a more natural task [for an exhaustive explanation of the procedure see the original paper *Mathôt et al., 2015*]. Briefly, in Experiments 1 and 2, sixteen and twenty-five participants, respectively, searched for small letters ('Z' or 'H') in natural scenes (from *Tkačik et al., 2011*). As in the saccade planning and preference tasks, gaze position and pupil size were recorded with an Eyelink 1000 (SR Research, Ontario, Canada) at 1000 Hz. Stimuli were presented (1280×1024) using OpenSesame (*Mathôt et al., 2012*) with the PsychoPy backend (*Peirce et al., 2019*).

In Experiment 2, auditory digits (0–9) were presented with an inter-digit interval of 1500 ms during search - note that Experiment 1 did not feature the auditory dual task. Crucially, participants either performed a dual task wherein the count of a specific digit was monitored throughout search, or a single task where the number stream was ignored. The single and dual conditions were blocked, and the sequence of these blocks was random across participants.

### Data processing and analyses
Pupil size was averaged per fixation and subsequently z-transformed per participant (*Mathôt et al., 2015*). Fixations with pupil sizes deviating more than 3*SD* from the mean (within a participant) and fixations positioned outside of the monitor were excluded to mitigate possible confounds (Exp. 1:

4.32%, Exp. 2: 9.75% discarded). 57,127 and 214,449 fixations were analyzed from Experiments 1 and 2, respectively. Fixations were classified into 36 bins based on their direction (bins consistent with the saccade planning and preference tasks).

To investigate if saccade costs predicted saccade preferences when searching in natural scenes, we analyzed all fixations from Experiment 1, and fixations from the single condition in Experiment 2. Next, we computed the average saccade preference map separately for each experiment by calculating the percentage of saccades in any of the 36 directions. Linear mixed-effects models were used to investigate whether this preference map predicted pupil size on a fixation-by-fixation basis in both experiments. We controlled for as many possible factors that are known to covary with pupil size in our model to control for them as much as possible to attempt to access the underlying saccade cost signal (*Mathôt et al., 2015*; Wilkinson notation: pupil size ~ saccade preferences + luminance + saliency + fixation number + trial number + x gaze coordinate + y gaze coordinate + saccade duration + fixation duration + saccade amplitude + (1 + saccade preferences|participant)).

To investigate if costly saccades were avoided in particular when the overall level of demand increased via the dual task, we analyzed data from Experiment 2, The percentages of saccades made into each direction for the single and dual conditions were calculated. We subtracted these averaged preference maps to obtain an adjustment map: this revealed how participants altered their saccade preferences under additional demand (*Figure 4e*). We predicted pupil size using the average adjustment map for each direction while again controlling for many possible confounding factors in the single condition using a linear mixed-effects model on a fixation-by-fixation basis (Wilkinson notation: pupil size ~ saccade adjustment + luminance + saliency + fixation number + trial number + x gaze coordinate + y gaze coordinate + saccade duration + fixation duration + saccade amplitude + (1 + saccade adjustment|participant)).

## Acknowledgements

This project has received funding from the European Research Council (ERC) under the European Union's Horizon 2020 research and innovation program (grant agreement n° 863732).

## Additional information

### Funding

| Funder | Grant reference number | Author |
|---|---|---|
| European Research Council | 10.3030/863732 | Damian Koevoet Stefan van der Stigchel |

The funders had no role in study design, data collection and interpretation, or the decision to submit the work for publication.

### Author contributions

Damian Koevoet, Conceptualization, Resources, Software, Formal analysis, Validation, Investigation, Visualization, Methodology, Writing – original draft, Writing – review and editing; Laura Van Zantwijk, Data curation, Formal analysis, Investigation, Methodology, Writing – review and editing; Marnix Naber, Supervision, Methodology, Writing – review and editing; Sebastiaan Mathôt, Resources, Data curation, Software, Writing – review and editing; Stefan van der Stigchel, Resources, Supervision, Funding acquisition, Project administration, Writing – review and editing; Christoph Strauch, Conceptualization, Resources, Formal analysis, Supervision, Validation, Investigation, Visualization, Methodology, Writing – original draft, Project administration, Writing – review and editing

### Author ORCIDs

Damian Koevoet (ID) https://orcid.org/0000-0002-9395-6524
Stefan van der Stigchel (ID) https://orcid.org/0000-0002-5918-3521
Christoph Strauch (ID) https://orcid.org/0000-0002-6380-8635

## Ethics

Participants provided written informed consent before taking part, and were awarded monetary compensation or course credits. The experimental procedure was approved by Utrecht University's ethical review board of the Faculty of Social Sciences (22-0635).

Reviewer #3 (Public review): https://doi.org/10.7554/eLife.97760.3.sa1
Author response https://doi.org/10.7554/eLife.97760.3.sa2

---

# Additional files

## Supplementary files

MDAR checklist

## Data availability

Data and analyses scripts to reproduce the results are available via the Open Science Framework: https://osf.io/n3ktm/. The original data from *Mathôt et al., 2015* are available at https://github.com/smathot/materials_for_P0010.5.

The following dataset was generated:

| Author(s) | Year | Dataset title | Dataset URL | Database and Identifier |
|---|---|---|---|---|
| Koevoet D, Strauch C | 2025 | Effort drives saccade selection | https://doi.org/10.17605/OSF.IO/N3KTM | Open Science Framework, 10.17605/OSF.IO/N3KTM |

The following previously published dataset was used:

| Author(s) | Year | Dataset title | Dataset URL | Database and Identifier |
|---|---|---|---|---|
| Mathôt S | 2014 | P0010.5 Cross-experimental analysis | https://github.com/smathot/materials_for_P0010.5 | GitHub, aef127f |

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

**Appendix 1**

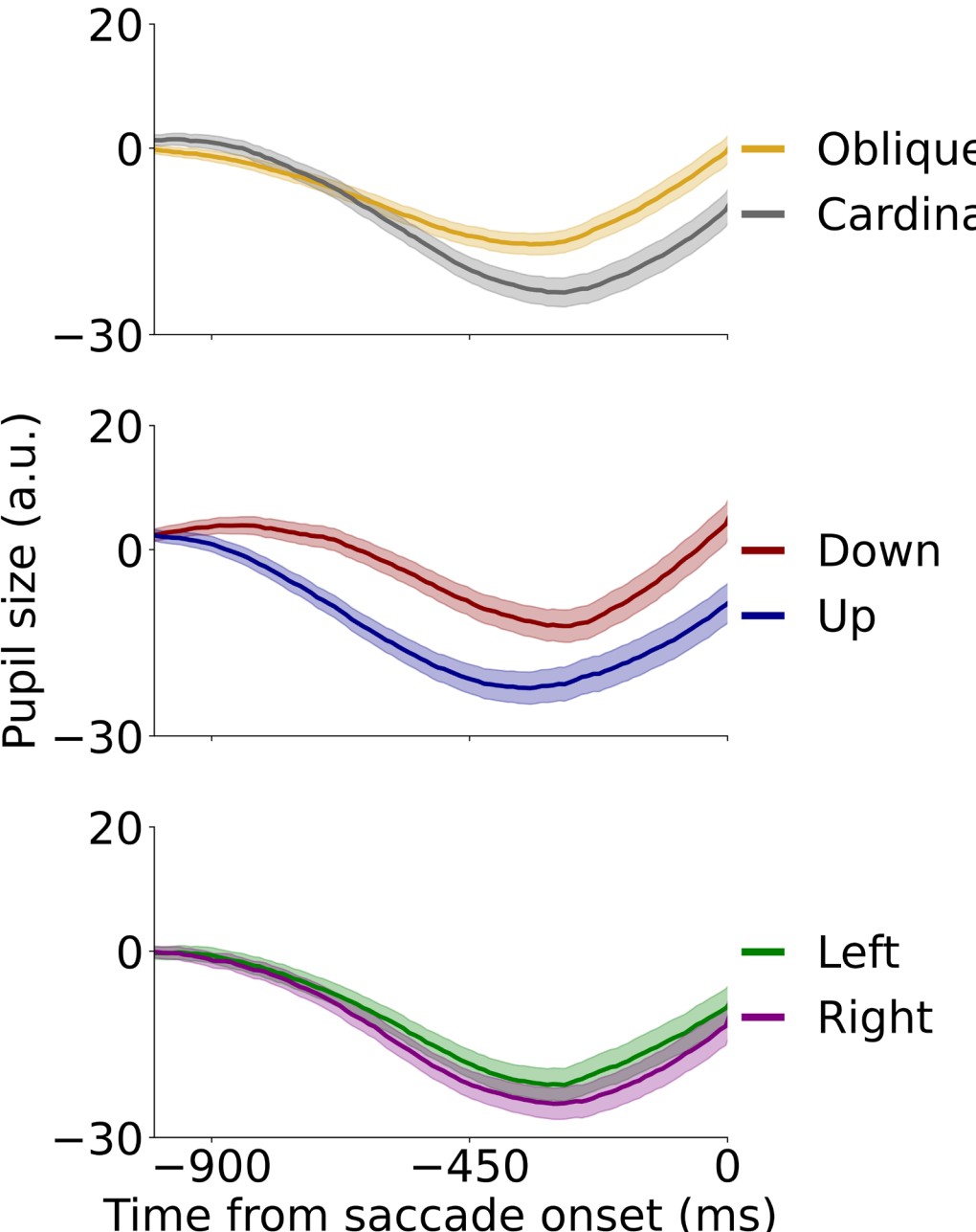

**Appendix 1—figure 1.** Saccade-locked pupil traces as index of saccade costs in different directions. Using the average pupil size in the 350 ms before saccade onset (saccade-locked), results remained qualitatively identical. Planning oblique saccades was associated with a larger pupil size than cardinal ones ($\beta = 9.897$, SE $= 2.223$, $t = 4.451$, $p < 0.001$), and downward saccades were associated with a larger pupil size than upward saccades ($\beta = 0.471$, SE $= 0.112$, $t = 4.189$, $p < 0.001$). A slightly increased pupil size for leftward compared with rightward saccades was observed as well ($\beta = 0.260$, SE $= 0.107$, $t = 2.436$, $p = 0.015$).

## Supporting analysis

To ascertain whether pupil size or other oculomotor metrics predict saccade preferences, we conducted a multiple regression analysis. We calculated average pupil size, saccade latency, landing precision, and peak velocity maps across all 36 directions. The model, determined using AIC-based

backward selection, included pupil size, latency, and landing precision as predictors (Wilkinson notation: saccade preferences ~ pupil size+ saccade latency+ landing precision). The analysis revealed that pupil size (β = –42.853, $t$ = 4.791, p < 0.001) and saccade latency (β = –0.377, $t$ = 2.106, p = 0.043) predicted saccade preferences. Landing precision did not reach significance (β = 23.631, $t$ = 1.675, p = 0.104). Together, this demonstrates that although other oculomotor metrics such as saccade latency contribute to saccade selection, pupil size remains a robust marker of saccade selection.

**Appendix 1—table 1.** Full outcomes of the linear mixed-effects model analyzing pupil size assessed saccade costs across directions.

| Predictor | β | SE | t | p |
|---|---|---|---|---|
| Intercept | –26.510 | 9.059 | –2.926 | 0.003 |
| Obliqueness | 7.662 | 1.957 | 3.916 | <0.001 |
| Verticalness | –0.556 | 0.171 | –3.261 | 0.001 |
| Horizontalness | –0.226 | 0.095 | –2.388 | 0.017 |
| Obliqueness × Duration | –0.109 | 0.031 | –3.495 | <0.001 |
| Duration | 0.169 | 0.085 | 1.981 | 0.048 |
| Amplitude | –2.423 | 0.656 | –3.691 | <0.001 |
| Landing error | 7.344 | 2.168 | 3.387 | 0.001 |
| Peak velocity | 0.054 | 0.015 | 3.692 | <0.001 |
| Participant Var | 602.851 | 2.786 | | |
| Participant × Obliqueness Cov | –55.890 | 0.395 | | |
| Participant × Verticalness Cov | –1.625 | 0.058 | | |
| Obliqueness × Verticalness Cov | 0.934 | 0.011 | | |
| Obliqueness Var | 13.790 | 0.083 | | |
| Verticalness Var | 0.380 | 0.003 | | |

**Appendix 1—table 2.** Full outcomes of the linear mixed-effects model predicting pupil size using saccade preferences and control variables in Experiment 1.

| Predictor | β | SE | t | p |
|---|---|---|---|---|
| Intercept | 0.825 | 0.061 | 13.504 | <0.001 |
| Direction preferences | –1.784 | 0.324 | –5.412 | <0.001 |
| X coordinate | –0.0002 | 0.00001 | –21.12 | <0.001 |
| Y coordinate | –0.0002 | 0.00001 | –11.316 | <0.001 |
| Luminance | –0.635 | 0.017 | –36.251 | <0.001 |
| Saliency | –0.001 | 0.0001 | –10.536 | <0.001 |
| Trial number | –0.006 | 0.00006 | –91.139 | <0.001 |
| Fixation number (in trial) | 0.006 | 0.0002 | 55.513 | <0.001 |
| Saccade duration | –0.004 | 0.0009 | –42.150 | <0.001 |
| Fixation duration | 0.0003 | 0.00004 | 7.105 | <0.001 |
| Amplitude | 0.009 | 0.0007 | 13.979 | <0.001 |
| Participant Var | 0.053 | | | |
| Direction preferences Var | 0.833 | 0.646 | | |
| Participant × Direction preferences Cov | 0.003 | 0.250 | | |

**Appendix 1—table 3.** Full outcomes of the linear mixed-effects model predicting pupil size using saccade preferences and control variables in Experiment 2.

| Predictor | β | SE | t | p |
|---|---|---|---|---|
| Intercept | 1.093 | 0.060 | 18.301 | <0.001 |
| Direction preferences | –0.644 | 0.170 | –3.780 | <0.001 |
| X coordinate | –0.0001 | 0.000005 | –24.442 | <0.001 |
| Y coordinate | –0.00004 | 0.000007 | –5.072 | <0.001 |
| Luminance | –0.341 | 0.009 | –38.269 | <0.001 |
| Saliency | –0.001 | 0.00006 | –10.405 | <0.001 |
| Trial number | –0.007 | 0.00006 | –126.687 | <0.001 |
| Fixation number (in trial) | 0.006 | 0.00009 | 64.174 | <0.001 |
| Saccade duration | –0.007 | 0.00005 | –152.775 | <0.001 |
| Fixation duration | 0.000005 | 0.00002 | 0.732 | 0.464 |
| Amplitude | 0.013 | 0.0003 | 37.737 | <0.001 |
| Participant Var | 0.086 | 0.041 | | |
| Direction preferences Var | 0.322 | 0.339 | | |
| Participant × Direction preferences Cov | 0.034 | 0.086 | | |

**Appendix 1—table 4.** Full outcomes of the linear mixed-effects model predicting pupil size using saccade direction adjustment and control variables.

| Predictor | β | SE | t | p |
|---|---|---|---|---|
| Intercept | 1.070 | 0.060 | 17.880 | <0.001 |
| Direction adjustment | –9.333 | 0.966 | –9.659 | <0.001 |
| X coordinate | –0.0001 | 0.000005 | –23.388 | <0.001 |
| Y coordinate | –0.00004 | 0.000007 | –5.261 | <0.001 |
| Luminance | –0.340 | 0.009 | –38.269 | <0.001 |
| Saliency | –0.001 | 0.00006 | –10.109 | <0.001 |
| Trial number | –0.007 | 0.00006 | –126.380 | <0.001 |
| Fixation number (in trial) | 0.006 | 0.00009 | 62.648 | <0.001 |
| Saccade duration | –0.007 | 0.00005 | –153.597 | <0.001 |
| Fixation duration | 0.00002 | 0.00002 | 1.537 | 0.124 |
| Amplitude | 0.012 | 0.0003 | 37.127 | <0.001 |
| Participant Var | 0.087 | 0.041 | | |
| Direction adjustment Var | 16.878 | 11.338 | | |
| Participant × Direction adjustment Cov | 0.142 | 0.499 | | |

