## [Editor Report · eLife Assessment]

This study provides **important** findings on the nature of eye movement choices by human subjects. The study uses a novel approach and provides relatively clear and **convincing** results of the relationship between pupil size and saccade production. The results should be of interest to a broad audience interested in sensorimotor integration and sensory-guided decision-making.

---

## [Referee Report · Reviewer #3 (Public review)]

Summary:

This manuscript extends previous research by this group by relating variation in pupil size to the endpoints of saccades produced by human participants under various conditions including trial-based choices between pairs of spots and search for small items in natural scenes. Based on the premise that pupil size is a reliable proxy of "effort", the authors conclude that less costly saccade targets are preferred. Finding that this preference was influenced by the performance of a non-visual, attention-demanding task, the authors conclude that a common source of effort animates gaze behavior and other cognitive tasks.

Strengths:

Strengths of the manuscript include the novelty of the approach, the clarity of the findings, and the community interest in the problem.

Weaknesses:

Enthusiasm for this manuscript is reduced by the following weaknesses:

(1) A relationship between pupil size and saccade production seems clear based on the authors' previous and current work. What is at issue is the interpretation. The authors test one, preferred hypothesis, and the narrative of the manuscript treats the hypothesis that pupil size is a proxy of effort as beyond dispute or question. The stated elements of their argument seem to go like this:

PROPOSITION 1: Pupil size varies systematically across task conditions, being larger when tasks are more demanding.

PROPOSITION 2: Pupil size is related to the locus coeruleus.

PROPOSITION 3: The locus coeruleus NE system modulates neural activity and interactions.

CONCLUSION: Therefore, pupil size indexes the resource demand or "effort" associated with task conditions.

How the conclusion follows from the propositions is not self-evident. Proposition 3, in particular, fails to establish the link that is supposed to lead to the conclusion.

(2) The authors test one, preferred hypothesis and do not consider plausible alternatives. Is "cost" the only conceivable hypothesis? The hypothesis is framed in very narrow terms. For example, the cholinergic and dopamine systems that have been featured in other researchers' consideration of pupil size modulation are missing here. Thus, because the authors do not rule out plausible alternative hypotheses, the logical structure of this manuscript can be criticized as committing the fallacy of affirming the consequent.

(3) The authors cite particular publications in support of the claim that saccade selection is influenced by an assessment of effort. Given the extensive work by others on this general topic, the skeptic could regard the theoretical perspective of this manuscript as too impoverished. Their work may be enhanced by consideration of other work on this general topic, e.g, (i) Shenhav A, Botvinick MM, Cohen JD. (2013) The expected value of control: an integrative theory of anterior cingulate cortex function. Neuron. 2013 Jul 24;79(2):217-40. (ii) Müller T, Husain M, Apps MAJ. (2022) Preferences for seeking effort or reward information bias the willingness to work. Sci Rep. 2022 Nov 14;12(1):19486. (iii) Bustamante LA, Oshinowo T, Lee JR, Tong E, Burton AR, Shenhav A, Cohen JD, Daw ND. (2023) Effort Foraging Task reveals a positive correlation between individual differences in the cost of cognitive and physical effort in humans. Proc Natl Acad Sci U S A. 2023 Dec 12;120(50):e2221510120.

(4) What is the source of cost in saccade production? What is the currency of that cost? The authors state (page 13), "... oblique saccades require more complex oculomotor programs than horizontal eye movements because more neuronal populations in the superior colliculus (SC) and frontal eye fields (FEF) [76-79], and more muscles are necessary to plan and execute the saccade [76, 80, 81]." This statement raises questions and concerns. First, the basis of the claim that more neurons in FEF and SC are needed for oblique versus cardinal saccades is not established in any of the publications cited. Second, the authors may be referring to the fact that oblique saccades require coordination between pontine and midbrain circuits. This must be clarified. Second, the cost is unlikely to originate in extraocular muscle fatigue because the muscle fibers are so different from skeletal muscles, being fundamentally less fatigable. Third, if net muscle contraction is the cost, then why are upward saccades, which require the eyelid, not more expensive than downward? Thus, just how some saccades are more effortful than others is not clear.

(5) The authors do not consider observations about variation in pupil size that seem to be incompatible with the preferred hypothesis. For example, at least two studies have described systematically larger pupil dilation associated with faster relative to accurate performance in manual and saccade tasks (e.g., Naber M, Murphy P. Pupillometric investigation into the speed-accuracy trade-off in a visuo-motor aiming task. Psychophysiology. 2020 Mar;57(3):e13499; Reppert TR, Heitz RP, Schall JD. Neural mechanisms for executive control of speed-accuracy trade-off. Cell Rep. 2023 Nov 28;42(11):113422). Is the fast relative to the accurate option necessarily more costly?

(6) The authors draw conclusions based on trends across participants, but they should be more transparent about variation that contradicts these trends. In Figures 3 and 4 we see many participants producing behavior unlike most others. Who are they? Why do they look so different? Is it just noise, or do different participants adopt different policies?

Comments on revisions:

The authors have addressed the concerns and questions raised in the original review.

---

## [Author Response]

The following is the authors’ response to the original reviews.

**Public Reviews:**

**Reviewer #1 (Public Review):**
Vision is a highly active process. Humans move their eyes 3-4 times per second to sample information with high visual acuity from our environment, and where eye movements are directed is critical to our understanding of active vision. Here, the authors propose that the cost of making a saccade contributes critically to saccade selection (i.e., whether and where to move the eyes). The authors build on their own recent work that the effort (as measured by pupil size) that comes with planning and generating an eye movement varies with saccade direction. To do this, the authors first measured pupil size for different saccade directions for each participant. They then correlated the variations in pupil size obtained in the mapping task with the saccade decision in a free-choice task. The authors observed a striking correlation: pupil size in the mapping task predicted the decision of where to move the eyes in the free choice task. In this study, the authors provide a number of additional insightful analyses (e.g., based on saccade curvature, and saccade latency) and experiments that further support their claim that the decision to move the eyes is influenced by the effort to move the eyes in a particular direction. One experiment showed that the same influence of assumed saccade costs on saccade selection is observed during visual search in natural scenes. Moreover, increasing the cognitive load by adding an auditory counting task reduced the number of saccades, and in particular reduced the costly saccades. In sum, these experiments form a nice package that convincingly establishes the association between pupil size and saccade selection.

We thank the reviewer for highlighting the novelty and cogency of our findings.

In my opinion, the causal structure underlying the observed results is not so clear. While the relationship between pupil size and saccade selection is compelling, it is not clear that saccade-related effort (i.e., the cost of a saccade) really drives saccade selection. Given the correlational nature of this relationship, there are other alternatives that could explain the finding. For example, saccade latency and the variance in landing positions also vary across saccade directions. This can be interpreted for instance that there are variations in oculomotor noise across saccade directions, and maybe the oculomotor system seeks to minimize that noise in a free-choice task. In fact, given such a correlational result, many other alternative mechanisms are possible. While I think the authors' approach of systematically exploring what we can learn about saccade selection using pupil size is interesting, it would be important to know what exactly pupil size can add that was not previously known by simply analyzing saccade latency. For example, saccade latency anisotropies across saccade directions are well known, and the authors also show here that saccade costs are related to saccade latency. An important question would be to compare how pupil size and saccade latency uniquely contribute to saccade selection. That is, the authors could apply the exact same logic to their analysis by first determining how saccade latencies (or variations in saccade landing positions; see Greenwood et al., 2017 PNAS) vary across saccade directions and how this saccade latency map explains saccade selection in subsequent tasks. Is it more advantageous to use one or the other saccade metric, and how well does a saccade latency map correlate with a pupil size map?

We thank the reviewer for the detailed comment. (1) The reviewer first points out the correlational nature of many of our results. Thereafter, (2), the reviewer asks whether saccade latencies and landing precision also predict saccade selection, and could be these potential predictors be considered alternative explanations to the idea of effort driving saccade selection? Moreover, what can pupil size add to what can be learned from saccade latency?

In brief, although we report a combination of correlational and causal findings, we do not know of a more parsimonious explanation for our findings than “effort drives saccade selection”. Moreover, we demonstrate that oculomotor noise cannot be construed as an alternative explanation for our findings.

(1) Correlational nature of many findings.

We acknowledge that many of our findings are predominantly correlational in nature. In our first tasks, we correlated pupil size during saccade planning to saccade preferences in a subsequent task. Although the link between across tasks was correlational, the observed relationship clearly followed our previously specified directed hypothesis. Moreover, experiments 1 and 2 of the visual search data replicated and extended this relationship. We also directly manipulated cognitive demand in the second visual search experiment. In line with the hypothesis that effort affects saccade selection, participants executed less saccades overall when performing a (primary) auditory dual task, and even cut the costly saccades most – which actually constitutes causal evidence for our hypothesis. A minimal oculomotor noise account would not directly predict a reduction in saccade rate under higher cognitive demand. To summarize, we have a combination of correlational and causal findings, although mediators cannot be ruled out fully for the latter. That said, we do not know of a more fitting and parsimonious explanation for our findings than effort predicting saccade selection (see following points for saccade latencies). We now address causality in the discussion for transparency and point more explicitly to the second visual search experiment for causal evidence.

“We report a combination of correlational and causal findings. Despite the correlational nature of some of our results, they consistently support the hypothesis that saccade costs predicts saccade selection [which we predicted previously, 33]. Causal evidence was provided by the dual-task experiment as saccade frequencies - and especially costly saccades were reduced under additional cognitive demand. Only a cost account predicts (1) a link between pupil size and saccade preferences, (2) a cardinal saccade bias, (3) reduced saccade frequency under additional cognitive demand, and (4) disproportional cutting of especially those directions associated with more pupil dilation. Together, our findings converge upon the conclusion that effort drives saccade selection.”

(2) Do anisotropies in saccade latencies constitute an alternative explanation?

First of all, we would like to to first stress that differences in saccade latencies are indeed thought to reflect oculomotor effort (Shadmehr et al., 2019; TINS). For example, saccades with larger amplitudes and saccades where distractors need to be ignored are associated with longer latencies. Therefore, even if saccade latencies would predict saccade selection, this would not contrast the idea that effort drives saccade selection. Instead, this would provide convergent evidence for our main novel conclusion: effort drives saccade selection. There are several reasons why pupil size can be used as a more general marker of effort (see responses to R2), but ultimately, our conclusions do not hinge on the employed measure of effort per se. As stressed above in (1), we see no equally parsimonious explanation besides the cost account. Moreover, we predicted this relationship in our previous publication before running the currently reported experiments and analyses (Koevoet et al., 2023). That said, we are open to discuss further alternative options and would be looking forward to test these accounts in future work against each other – we are welcoming the reviewers’ (but also the reader’s) suggestions.

We now discuss this in the manuscript as follows:

“We here measured cost as the degree of effort-linked pupil dilation. In addition to pupil size, other markers may also indicate saccade costs. For example, saccade latency has been proposed to index oculomotor effort [100], whereby saccades with longer latencies are associated with more oculomotor effort. This makes saccade latency a possible complementary marker of saccade costs (also see Supplemen- tary Materials). Although relatively sluggish, pupil size is a valuable measure of attentional costs for (at least) two reasons. First, pupil size is a highly established as marker of effort, and is sensitive to effort more broadly than only in the context of saccades [36–45, 48]. Pupil size therefore allows to capture not only the costs of saccades, but also of covert attentional shifts [33], or shifts with other effectors such as head or arm movements [54, 101]. Second, as we have demonstrated, pupil size can measure saccade costs even when searching in natural scenes (Figure 4). During natural viewing, it is difficult to disentangle fixation duration from saccade latencies, complicating the use of saccade latency as a measure of saccade cost.

Together, pupil size, saccade latency, and potential other markers of saccade cost could fulfill complementary roles in studying the role of cost in saccade selection.”

Second, we followed the reviewer’s recommendation in testing whether other oculomotor metrics would predict saccade selection. To this end, we conducted a linear regression across directions. We calculated pupil size, saccade latencies, landing precision and peak velocities maps from the saccade planning task. We then used AICbased backward model selection to determine the ‘best’ model model to determine which factor would predict saccade selection best. The best model included pupil size, latency and landing precision as predictors (Wilkinson notation: saccade preferences ~ pupil size + saccade latency + landing precision). Pupil size (b = -42.853, *t* = 4.791, *p* < .001) and saccade latency (b = -.377, *t* = 2.106, *p* = .043; see Author response image 1) predicted saccade preferences significantly. In contrast, landing precision did not reach significance (b = 23.631, *t* = 1.675, *p* = .104). This analysis shows that although saccade latency also predicts saccade preferences, pupil size remains a robust predictor of saccade selection. These findings demonstrate that minimizing oculomotor noise cannot fully explain the pattern of results.

**Author response image 1. sa2fig1:** The relationship between saccade latency (from the saccade planning task) and saccade preferences averaged across participants. Individual points reflect directions and shading represents bootstrapped 95% confidence intervals.

We have added this argument into the manuscript, and discuss the analysis in the discussion. Details of the analysis have been added to the Supporting Information for transparency and further detail.

“A control analysis ruled out that the correlation between pupil size and saccade preferences was driven by other oculomotor metrics such as saccade latency and landing precision (see Supporting Information).”

“To ascertain whether pupil size or other oculomotor metrics predict saccade preferences, we conducted a multiple regression analysis. We calculated average pupil size, saccade latency, landing precision and peak velocity maps across all 36 directions. The model, determined using AIC-based backward selection, included pupil size, latency and landing precision as predictors (Wilkinson notation: saccade preferences pupil size + saccade latency + landing precision). The analysis re- vealed that pupil size (β = -42.853, t = 4.791, p < .001) and saccade latency (β = -.377, t = 2.106, p = .043) predicted saccade preferences. Landing precision did not reach significance (β = 23.631, t = 1.675, p = .104). Together, this demonstrates that although other oculomotor metrics such as saccade latency contribute to saccade selection, pupil size remains a robust marker of saccade selection.”

In addition to eye-movement-related anisotropies across the visual field, there are of course many studies reporting visual field anisotropies (see Himmelberg, Winawer & Carrasco, 2023, Trends in Neuroscience for a review). It would be interesting to understand how the authors think about visual field anisotropies in the context of their own study. Do they think that their results are (in)dependent on such visual field variations (see Greenwood et al., 2017, PNAS; Ohl, Kroell, & Rolfs, 2024, JEP:Gen for a similar discussion)?

We agree that established visual field anisotropies are fascinating to be discussed in context of our own results. At the reviewer’s suggestion, we now expanded this discussion.

The observed anisotropies in terms of saccade costs are likely related to established anisotropies in perception and early visual cortex. However, the exact way that these anisotropies may be linked remains elusive (i.e. what is cause, what is effect, are links causal?), and more research is necessary to understand how these are related.

“The observed differences in saccade costs across directions could be linked to established anisotropies in perception [80–86], attention [87–92], saccade charac- teristics [87, 88, 92, 93], and (early) visual cortex [94–98] [also see 99]. For example, downward saccades are more costly than upward saccades, which mimics a similar asymmetry in early visual areas wherein the upper visual field is relatively under- represented [94–98]; similarly stronger presaccadic benefits are found for down- compared with upward saccades [87, 88]. Moreover, upward saccades are more pre- cise than downward saccades [93]. Future work should elucidate where saccade cost or the aforementioned anisotropies originate from and how they are related - something that pupil size alone cannot address.”

We also added that the finding that more precise saccades are coupled with worse performance in a crowding task might be attributed to the increased effort associated with more precise saccades (Greenwood et al., 2017).

“Adaptive resource allocation from, and to the oculomotor system parsimoniously explains a number of empirical observations. For example, higher cognitive demand is accompanied by smooth pursuits deviating more from to-be tracked targets [137], reduced (micro)saccade frequencies [Figure 4; 63, 64, 138, 139], and slower peak saccade velocities [140–142]. Relatedly, more precise saccades are accompanied with worse performance in a crowding task [93].”

Finally, the authors conclude that their results "suggests that the eye-movement system and other cognitive operations consume similar resources that are flexibly allocated among each other as cognitive demand changes. The authors should speculate what these similar resources could mean? What are the specific operations of the auditory task that overlap in terms of resources with the eye movement system?

We agree that the nature of joint resources is an interesting question. Our previous discussion was likely too simplistic here (see also responses to R3). We here specifically refer to the cognitive resources that one can flexibly distribute between tasks.

Our data do not directly speak to the question of what the shared resources between the auditory and oculomotor tasks are. Nevertheless, both tasks charge working memory as saccade targets are mandatorily encoded into working memory prior to saccade onset (Van der Stigchel & Hollingworth, 2018), and the counting task clearly engages working memory. This may indicate some domain-generality between visual and auditory working memory during natural viewing (see Nozari & Martin, 2024 for a recent review), but this remains speculative. Another possibility is that not the working memory encoding associated with saccades per se, but that the execution of overt motor actions itself also requires cognitive processing as suggested by Beatty (1982): “the organization of an overt motor act places additional demands on informationprocessing resources that are reflected in the task-evoked pupillary response”.

We have added upon this in more detail in the results and discussion sections.

“Besides the costs of increased neural activity when exerting more effort, effort should be considered costly for a second reason: Cognitive resources are limited. Therefore, any unnecessary resource expenditure reduces cognitive and behavioral flexibility [22, 31, 36, 116]. As a result, the brain needs to distribute resources between cognitive operations and the oculomotor system. We found evidence for the idea that such resource distribution is adaptive to the general level of cognitive demand and available resources: Increasing cognitive demand through an additional pri- mary auditory dual task led to a lower saccade frequency, and especially costly sac- cades were cut. In this case, it is important to consider that the auditory task was the primary task, which should cause participants to distribute resources from the ocu- lomotor system to the counting task. In other situations, more resources could be distributed to the oculomotor system instead, for example to discover new sources of reward [22, 136]. Adaptive resource allocation from, and to the oculomotor system parsimoniously explains a number of empirical observations. For example, higher cognitive demand is accompanied by smooth pursuits deviating more from to-be tracked targets [137], reduced (micro)saccade frequencies [Figure 4; 63, 64, 138, 139], and slower peak saccade velocities [140–142]. Relatedly, more precise saccades are accompanied with worse performance in a crowding task [93]. Furthermore, it has been proposed that saccade costs are weighed against other cognitive operations such as using working memory [33, 143–146]. How would the resources between the oculomotor system and cognitive tasks (like the auditory counting task) be related? One possibility is that both consume from limited working memory resources [147, 148]. Saccades are thought to encode target objects in a mandatory fashion into (vi- sual) working memory [79], and the counting task requires participants to keep track of the auditory stream and maintain count of the instructed digit in working mem- ory. However, the exact nature of which resources overlap between tasks remain open for future investigation [also see 149]. Together, we propose that cognitive re- sources are flexibly (dis)allocated to and from the oculomotor system based on the current demands to establish an optimal balance between performance and cost minimization.”

**Reviewer #2 (Public Review):**
The authors attempt to establish presaccadic pupil size as an index of 'saccade effort' and propose this index as one new predictor of saccade target selection. They only partially achieved their aim: When choosing between two saccade directions, the less costly direction, according to preceding pupil size, is preferred. However, the claim that with increased cognitive demand participants would especially cut costly directions is not supported by the data. I would have expected to see a negative correlation between saccade effort and saccade direction 'change' under increased load. Yet participants mostly cut upwards saccades, but not other directions that, according to pupil size, are equally or even more costly (e.g. oblique saccades).Strengths:The paper is well-written, easy to understand, and nicely illustrated.The sample size seems appropriate, and the data were collected and analyzed using solid and validated methodology.Overall, I find the topic of investigating factors that drive saccade choices highly interesting and relevant.

We thank the reviewer for pointing out the strengths of our paper.

Weaknesses:The authors obtain pupil size and saccade preference measures in two separate tasks. Relating these two measures is problematic because the computations that underly saccade preparation differ. In Experiment 1, the saccade is cued centrally, and has to be delayed until a "go-signal" is presented; In Experiment 2, an immediate saccade is executed to an exogenously cued peripheral target. The 'costs' in Experiment 1 (computing the saccade target location from a central cue; withholding the saccade) do not relate to Experiment 2. It is unfortunate, that measuring presaccadic pupil size directly in the comparatively more 'natural' Experiment 2 (where saccades did not have to be artificially withheld) does not seem to be possible. This questions the practical application of pupil size as an index of saccade effort

This is an important point raised by the reviewer and we agree that a discussion on these points improves the manuscript. We reply in two parts: (1) Although the underlying computations during saccade preparation might differ, and are therefore unlikely to be fully similar (we agree), we can still predict saccade selection between (Saccade planning to Saccade preference) and within tasks (Visual search). (2) Pupil size is a sluggish physiological signal, but this is outweighed by the advantages of using pupil size as a general marker of effort, also in the context of visual selection compared with saccade latencies.

(1) Are delayed saccades (cost task) and the much faster saccades (preference task) linked?

As the reviewer notes the underlying ‘type’ of oculomotor program may differ between voluntarily delayed-saccades and those in the saccade preference task. There are, however, also considerable overlaps between the oculomotor programs as the directions and amplitudes are identical. Moreover, the different types of saccades have considerable overlap in their underlying neural circuitry. Nevertheless, the underlying oculomotor programs likely still differ in some regard. Even despite these differences, we were able to measure differences across directions in both tasks, and costs and preferences were negatively and highly correlated between tasks. The finding itself therefore indicates that the costs of saccades measured during the saccade planning task generalize to those in the saccade preference task. Note also that we predicted this finding and idea already in a previous publication before starting the present study (Koevoet et al., 2023).

We now address this interesting point in the discussion as follows:

“We observed that aOordable saccades were preferred over costly ones. This is especially remarkable given that the delayed saccades in the planning task likely differ in their oculomotor program from the immediate saccades in the preference task in some regard.”

(2) Is pupil size a sensible measure of saccade effort?

As the reviewer points out, the pupillary signal is indeed relatively sluggish and therefore relatively slow and more artifical tasks are preferred to quantify saccade costs. This does not preclude pupil size from being applied in more natural settings, as we demonstrate in the search experiments – but a lot of care has to be taken to control for many possible confounding factors and many trials will be needed.

That said, as saccade latencies may also capture differences in oculomotor effort (Shadmehr et al., 2019) they are a possible alternative option to assess effort in some oculomotor tasks (see below on why saccade latencies do not provide evidence for an alternative to effort driving saccade selection, but converging evidence). Whilst we do maintain that pupil size is an established and versatile physiological marker of effort, saccade latencies provide converging evidence for our conclusion that effort drives saccade selection.

As for the saccade preference task, we are not able to analyze the data in a similar manner as in the visual search task for two reasons. First, the number of saccades is much lower than in the natural search experiments. Second, in the saccade preference task, there were always two possible saccade targets. Therefore, even if we were able to isolate an effort signal, this signal could index a multitude of factors such as deciding between two possible saccade targets. Even simple binary decisions go hand in hand with reliable pupil dilations as they require effort (e.g. de Gee et al., 2014).

There are three major reasons why pupil size is a more versatile marker of saccade costs than saccade latencies (although as mentioned, latencies may constitute another valuable tool to study oculomotor effort). First, pupil size is able to quantify the cost of attentional shifts more generally, including covert attention as well as other effector systems such as head and hand movements. This circumvents the issue of different latencies of different effector systems and also allows to study attentional processes that are not associated with overt motor movements. Second, saccade latencies are difficult to interpret in natural viewing data, as fixation duration and saccade latencies are inherently confounded by one another. This makes it very difficult to separate oculomotor processes and the extraction of perceptual information from a fixated target. Thus, pupil size is a versatile marker of attentional costs in a variety of settings, and can measure costs that saccade latencies cannot (i.e. covert attention). Lastly, pupil size is highly established as a marker of effort which has been demonstrated across wide range of cognitive tasks and therefore not bound to eye movements alone (Bumke, 1911; Koevoet et al., 2024; Laeng et al., 2012; Loewenfeld, 1958; Mathôt, 2018; Robison & Unsworth, 2019; Sirois & Brisson, 2014; Strauch et al., 2022; van der Wel & van Steenbergen, 2018).

We now discuss this as follows:

“We here measured cost as the degree of effort-linked pupil dilation. In addition to pupil size, other markers may also indicate saccade costs. For example, saccade latency has been proposed to index oculomotor effort [100], whereby saccades with longer latencies are associated with more oculomotor effort. This makes saccade latency a possible complementary marker of saccade costs (also see Supplemen- tary Materials). Although relatively sluggish, pupil size is a valuable measure of attentional costs for (at least) two reasons. First, pupil size is a highly established as marker of effort, and is sensitive to effort more broadly than only in the context of saccades [36–45, 48]. Pupil size therefore allows to capture not only the costs of saccades, but also of covert attentional shifts [33], or shifts with other effectors such as head or arm movements [54, 101]. Second, as we have demonstrated, pupil size can measure saccade costs even when searching in natural scenes (Figure 4). During natural viewing, it is difficult to disentangle fixation duration from saccade latencies, complicating the use of saccade latency as a measure of saccade cost. Together, pupil size, saccade latency, and potential other markers of saccade cost could fulfill complementary roles in studying the role of cost in saccade selection.”

The authors claim that the observed direction-specific 'saccade costs' obtained in Experiment 1 "were not mediated by differences in saccade properties, such as duration, amplitude, peak velocity, and landing precision (Figure 1e,f)". Saccade latency, however, was not taken into account here but is discussed for Experiment 2.

The final model that was used to test for the observed anisotropies in pupil size across directions indeed did not include saccade latencies as a predictor. However, we did consider saccade latencies as a potential predictor originally. As we performed AICbased backward model selection, however, this predictor was removed due to the marginal predictive contribution of saccade latency beyond other predictors explaining pupil size.

For completeness, we here report the outcome of a linear mixed-effects that does include saccade latency as a predictor. Here, saccade latencies did not predict pupil size (b = 1.859e-03, *t* = .138, *p* = .889). The asymmetry effects remained qualitatively unchanged: preparing oblique compared with cardinal saccades resulted in a larger pupil size (b = 7.635, *t* = 3.969, *p* < .001), and preparing downward compared with upward saccades also led to a larger pupil size (b = 3.344, *t* = 3.334, *p* = .003).

The apparent similarity of saccade latencies and pupil size, however, is striking. Previous work shows shorter latencies for cardinal than oblique saccades, and shorter latencies for horizontal and upward saccades than downward saccades - directly reflecting the pupil sizes obtained in Experiment 1 as well as in the authors' previous study (Koevoet et al., 2023, PsychScience).

As the reviewer notes, there are substantial asymmetries across the visual field in saccade latencies. These assymetries in saccade latency could also predict saccade preferences. We will reply to this in three points: (1) even if saccade latency is a predictor of saccade preferences, this would not constitute as an alternative explanation to the conclusion of effort driving saccade selection, (2) saccade latencies show an up-down asymmetry but oblique-cardinal effects in latency may not be generalizable across saccade tasks, (3) pupil size remains a robust predictor of saccade preferences even when saccade latencies are considered as a predictor of saccade preferences.

(1) We want to first stress that saccade latencies are thought to reflect oculomotor effort (Shadmehr et al., 2019). For example, saccades with larger amplitudes and saccades where distractors need to be ignored are associated with longer latencies. Therefore, even if saccade latencies predict saccade selection, this would not contrast the idea that effort drives saccade selection. Instead, this would provide convergent evidence for our main conclusion – effort predicting saccade selection (rather than pupil size predicting saccade selection per se).

“We here measured cost as the degree of effort-linked pupil dilation. In addition to pupil size, other markers may also indicate saccade costs. For example, saccade latency has been proposed to index oculomotor effort [100], whereby saccades with longer latencies are associated with more oculomotor effort. This makes saccade latency a possible complementary marker of saccade costs (also see Supplemen- tary Materials). Although relatively sluggish, pupil size is a valuable measure of attentional costs for (at least) two reasons. First, pupil size is a highly established as marker of effort, and is sensitive to effort more broadly than only in the context of saccades [36–45, 48]. Pupil size therefore allows to capture not only the costs of saccades, but also of covert attentional shifts [33], or shifts with other effectors such as head or arm movements [54, 101]. Second, as we have demonstrated, pupil size can measure saccade costs even when searching in natural scenes (Figure 4). During natural viewing, it is difficult to disentangle fixation duration from saccade latencies, complicating the use of saccade latency as a measure of saccade cost. Together, pupil size, saccade latency, and potential other markers of saccade cost could fulfill complementary roles in studying the role of cost in saccade selection.”

(2) We first tested anisotropies in saccade latency in the saccade planning task (Wilkinson notation: latency ~ obliqueness + updownness + leftrightness + saccade duration + saccade amplitude + saccade velocity + landing error + (1+obliqueness + updownness|participant)). We found upward latencies to be shorter than downward saccade latencies (b = -.535, *t* = 3.421, *p* = .003). In addition, oblique saccades showed shorter latencies than cardinal saccades (b = -1.083, *t* = 3.096, *p* = .002) – the opposite of what previous work has demonstrated.

We then also tested these latency anisotropies in another dataset wherein participants (*n* = 20) saccaded toward a single peripheral target as fast as possible (Koevoet et al., submitted; same amplitude and eccentricity as in the present manuscript). There we did not find a difference in saccade latency between cardinal and oblique targets, but we did observe shorter latencies for up- compared with downward saccades. We are therefore not sure in which situations oblique saccades do, or do not differ from cardinal saccades in terms of latency, and even in which direction the effect occurs.

In contrast, we have now demonstrated a larger pupil size prior to oblique compared with cardinal saccades in two experiments. This indicates that pupil size may be a more reliable and generalizable marker of saccade costs than saccade latency. However, this remains to be investigated further.

(3) To gain further insights into which oculomotor metrics would predict saccade selection, we conducted a linear regression across directions. We created pupil size, saccade latencies, landing precision and peak velocities maps from the saccade planning task. We then used AIC-based model selection to determine the ‘best’ model to determine which factor would predict saccade selection best. The selected model included pupil size, latency and landing precision as predictors (Wilkinson notation: saccade preferences ~ pupil size + saccade latency + landing precision). Pupil size (b = -42.853, *t* = 4.791, *p* < .001) and saccade latency (b = -.377, *t* = 2.106, *p* = .043) predicted saccade preferences significantly. In contrast, landing precision did not reach significance (b = 23.631, *t* = 1.675, *p* = .104). This analysis shows that although saccade latency predicts saccade preferences, pupil size remains a robust predictor of saccade selection.

“To ascertain whether pupil size or other oculomotor metrics predict saccade preferences, we conducted a multiple regression analysis. We calculated average pupil size, saccade latency, landing precision and peak velocity maps across all 36 directions. The model, determined using AIC-based backward selection, included pupil size, latency and landing precision as predictors (Wilkinson notation: saccade preferences pupil size + saccade latency + landing precision). The analysis re- vealed that pupil size (β = -42.853, t = 4.791, p < .001) and saccade latency (β = -.377, t = 2.106, p = .043) predicted saccade preferences. Landing precision did not reach significance (β = 23.631, t = 1.675, p = .104). Together, this demonstrates that although other oculomotor metrics such as saccade latency contribute to saccade selection, pupil size remains a robust marker of saccade selection.”

The authors state that "from a costs-perspective, it should be eOicient to not only adjust the number of saccades (non-specific), but also by cutting especially expensive directions the most (specific)". However, saccade targets should be selected based on the maximum expected information gain. If cognitive load increases (due to an additional task) an effective strategy seems to be to perform less - but still meaningful - saccades. How would it help natural orienting to selectively cut saccades in certain (effortful) directions? Choosing saccade targets based on comfort, over information gain, would result in overall more saccades to be made - which is non-optimal, also from a cost perspective.

We thank the reviewer for this comment. Although we do not fully agree, the logic is quite close to our rationale and it is worth adding a point of discussion here. A vital part of the current interpretation is the instruction given to participants. In our second natural visual search task, participants were performing a dual task, where the auditory task was the primary task, whilst the search task was secondary. Therefore, participants are likely to adjust their resources to optimize performance on the primary task – at the expense of the secondary task. Therefore, less resources are made available and used to searching in the dual than in the single task, because these resources are needed for the auditory task. Cutting expensive directions does not help search in terms of search performance, but it does reduce the cost of search, so that more resources are available for the prioritized auditory task. Also note that the search task was rather difficult – participants did it, but it was tough (see the original description of the dataset for more details), which provides another reason to go full in on the auditory task at expense of the visual task. This, however, opens up a nice point of discussion: If one would emphasize the importance of search (maybe with punishment or reward), we would indeed expect participants to perform whichever eye movements are getting them to their goal fastest – thus reducing the relative influence of costs on saccade behavior. This remains to be tested however - we are working on this and are looking forward to discussing such findings in the future.

Together, we propose that there is a trade-off between distributing resources either towards cognitive tasks or the oculomotor system (also see Ballard et al., 1995; Van der Stigchel, 2020). How these resources are distributed depends highly on the current task demands (also see Sahakian et al., 2023). This allows for adaptive behavior in a wide range of contexts.

We now added these considerations to the manuscript as follows (also see our previous replies):

“Do cognitive operations and eye movements consume from a similar pool of resources [44]? If so, increasing cognitive demand for non-oculomotor processes should result in decreasing available resources for the oculomotor system. In line with this idea, previous work indeed shows altered eye-movement behavior un- der effort as induced by dual tasks, for example by making less saccades under increased cognitive demand [62–64]. We therefore investigated whether less sac- cades were made as soon as participants had to count the occurrence of a specific digit in the auditory number stream in comparison to ignoring the stream (in Exp. 2; Figure 4a). Participants were instructed to prioritize the auditory digit-counting task over finding the visual search target. Therefore, resources should be shifted from the oculomotor system to the primary auditory counting task. The additional cognitive demand of the dual task indeed led to a decreased saccade frequency (t(24) = 7.224, p < .001, Cohen’s d = 1.445; Figure 4h).”

I would have expected to see a negative correlation between saccade effort and saccade direction 'change' under increased load. Yet participants mostly cut upwards saccades, but not other directions that, according to pupil size, are equally or even more costly (e.g. oblique saccades).

The reviewer’s point is taken from the initial comment, which we will address here. First, we’d like to point out that is it not established that saccade costs in different directions are always the same. Instead, it is possible that saccade costs could be different in natural viewing compared with our delayed-saccade task. Therefore, we used pupil size during natural viewing for the search experiments. Second, the reviewer correctly notes that oblique saccades are hardly cut when under additional cognitive demand. However, participants already hardly execute oblique saccades when not confronted with the additional auditory task (Figure 4b, d), making it difficult to reduce those further (i.e. floor effect). Participants chose to cut vertical saccades, possibly because these are more costly than horizontal saccades.

We incorporated these point in our manuscript as follows:

“To test this, we analyzed data from two existing datasets [63] wherein participants (total *n* = 41) searched for small targets (’Z’ or ’H’) in natural scenes (Figure 4a; [64]). Again, we tested whether pupil size prior to saccades negatively linked with saccade preferences across directions. Because saccade costs and preferences across directions could differ for different situations (i.e. natural viewing vs. saccade preference task), but should always be negatively linked, we established both cost and preferences independently in each dataset.”

“We calculated a saccade-adjustment map (Figure 4g) by subtracting the saccade preference map in the single task (Figure 4f) from the dual task map (Fig- ure 4d). Participants seemingly cut vertical saccades in particular, and made more saccades to the top right direction. This pattern may have emerged as vertical saccades are more costly than horizontal saccades (also see Figure 1d). Oblique saccades may not have been cut because there were very little oblique saccades in the single condition to begin with (Figure 4d), making it difficult to observe a further reduction of such saccades under additional cognitive demand (i.e. a floor effect).”

Overall, I am not sure what practical relevance the relation between pupil size (measured in a separate experiment) and saccade decisions has for eye movement research/vision science. Pupil size does not seem to be a straightforward measure of saccade effort. Saccade latency, instead, can be easily extracted in any eye movement experiment (no need to conduct a separate, delayed saccade task to measure pupil dilation), and seems to be an equally good index.

There are two points here.

(1) What is the practical relevance of a link between effort and saccade selection for eyemovement research and vision science?

We see plenty – think of changing eye movement patterns under effort (be it smooth pursuits, saccade rates, distributions of gaze positions to images etc.) which have substantial implications for human factors research, but also neuropsychology. With a cost account, one may predict (rather than just observe) how eye movement changes as soon as resources are reduced/ non-visual demand increases. With a cost account, we can explain such effects (e.g. lower saccade rates under effort, cardinal bias, perhaps also central bias) parsimoniously that cannot be explained by what is so far referred to as the three core drivers of eye movement behavior (saliency, selection history, goals, e.g., Awh et al., 2012). Conversely, one must wonder why eye-movement research/vision science simply accepts/dismisses these phenomena as such, without seeking overarching explanations.

(2) What is the usefulness of using pupil size to measure effort?

We hope that our replies to the comments above illustrate why pupil size is a sensible, robust and versatile marker of attentional costs. We briefly summarize our most important points here.

- Pupil size is an established measure of effort irrespective of context, as demonstrated by hundreds of original works (e.g. working memory load, multiple object tracking, individual differences in cognitive ability). This allows pupil size to be a versatile marker of the effort, and therefore costs, of non-saccadic attentional shifts such as covert attention or those realized by other effector systems (i.e. head or hand movements).

- Our new analysis indicates that pupil size remains a strong and robust predictor of saccade preference, even when considering saccade latency.

- Pupil size allows to study saccade costs in natural viewing. In contrast, saccade latencies are difficult to assess in natural viewing as fixation durations and saccade latencies are intrinsically linked and very difficult to disentangle.

- Note however, that we think that it is interesting and useful so study effects of effort/cost on eye movement behavior. Whichever index is used to do so, we see plenty potential in this line of research, this paper is a starting point to do so.

**Reviewer #3 (Public Review):**
This manuscript extends previous research by this group by relating variation in pupil size to the endpoints of saccades produced by human participants under various conditions including trial-based choices between pairs of spots and search for small items in natural scenes. Based on the premise that pupil size is a reliable proxy of "effort", the authors conclude that less costly saccade targets are preferred. Finding that this preference was influenced by the performance of a non-visual, attentiondemanding task, the authors conclude that a common source of effort animates gaze behavior and other cognitive tasks.Strengths:Strengths of the manuscript include the novelty of the approach, the clarity of the findings, and the community interest in the problem.

We thank the reviewer for pointing out the strengths of our paper.

Weaknesses:Enthusiasm for this manuscript is reduced by the following weaknesses:(1) A relationship between pupil size and saccade production seems clear based on the authors' previous and current work. What is at issue is the interpretation. The authors test one, preferred hypothesis, and the narrative of the manuscript treats the hypothesis that pupil size is a proxy of effort as beyond dispute or question. The stated elements of their argument seem to go like this:PROPOSITION 1: Pupil size varies systematically across task conditions, being larger when tasks are more demanding.PROPOSITION 2: Pupil size is related to the locus coeruleus.PROPOSITION 3: The locus coeruleus NE system modulates neural activity and interactions.CONCLUSION: Therefore, pupil size indexes the resource demand or "effort" associated with task conditions.How the conclusion follows from the propositions is not self-evident. Proposition 3, in particular, fails to establish the link that is supposed to lead to the conclusion.

We inadvertently laid out this rationale as described above, and we thank the reviewer for pointing out this initial suboptimal structure of argumentation. The notion that the link between pupil size and effort is established in the literature *because* of its neural underpinnings is inaccurate. Instead, the tight link between effort and pupil size is established based on covariations of pupil diameter and cognition across a wide variety of tasks and domains. In line with this, we now introduce this tight link predominantly based on the relationships between pupil size and cognition instead of focusing on putative neural correlates of this relationship.

As reviewed previously (Beatty, 1982; Bumke, 1911; Kahneman, 1973; Kahneman & Beatty, 1966; Koevoet et al., 2024; Laeng et al., 2012; Mathôt, 2018; Sirois & Brisson, 2014; Strauch et al., 2022; van der Wel & van Steenbergen, 2018), any increase in effort is consistently associated with an increase in pupil size. For instance, the pupil dilates when increasing load in working memory or multiple object tracking tasks, and such pupillary effects robustly explain individual differences in cognitive ability and fluctuations in performance across trials (Alnæs et al., 2014; Koevoet et al., 2024; Robison & Brewer, 2020; Robison & Unsworth, 2019; Unsworth & Miller, 2021). This extends to the planning of movements as pupil dilations are observed prior to the execution of (eye) movements (Koevoet et al., 2023; Richer & Beatty, 1985). The link between pupil size and effort has thus been firmly established for a long time, irrespective of the neural correlates of these effort-linked pupil size changes.

We again thank the reviewer for spotting this logical mistake, and now revised the paragraph where we introduce pupil size as an established marker of effort as follows:

“We recently demonstrated that the effort of saccade planning can be measured with pupil size, which allows for a physiological quantification of saccade costs as long as low-level visual factors are controlled for [33]. Pupil size is an established marker of effort [36–44]. For instance, loading more in working memory or tracking more objects results in stronger pupil dilation [44–52]. Pupil size not only reflects cognitive (or mental) effort but also the effort of planning and executing movements [37, 53, 54]. We leveraged this to demonstrate that saccade costs can be captured with pupil size, and are higher for oblique compared with cardinal directions [33]. Here, we addressed whether saccade costs predict where to saccade.”

We now mention the neural correlates of pupil size only in the discussion. Where we took care to also mention roles for other neurotransmitter systems:

“Throughout this paper, we have used cost in the limited context of saccades.

However, cost-based decision-making may be a more general property of the brain [31, 36, 114–116]. Every action, be it physical or cognitive, is associated with an in- trinsic cost, and pupil size is likely a general marker of this [44]. Note, however, that pupil dilation does not always reflect cost, as the pupil dilates in response to many sensory and cognitive factors which should be controlled for, or at least considered, when interpreting pupillometric data [e.g., see 39, 40, 42, 117]. Effort-linked pupil dilations are thought to be, at least in part, driven by activity in the brainstem locus coeruleus (LC) [40, 118–120] [but other neurotransmitters also affect pupil size, e.g. 121, 122]. Activity in LC with its widespread connections throughout the brain [120, 123–127] is considered to be crucial for the communication within and between neu- ral populations and modulates global neural gain [128–132]. Neural firing is costly [22, 133], and therefore LC activity and pupil size are (neuro)physiologically plausible markers of cost [40]. Tentative evidence even suggests that continued exertion of effort (accompanied by altered pupil dilation) is linked to the accumulation of glutamate in the lateral prefrontal cortex [134], which may be a metabolic marker of cost [also see 116, 134, 135]. “

(2) The authors test one, preferred hypothesis and do not consider plausible alternatives. Is "cost" the only conceivable hypothesis? The hypothesis is framed in very narrow terms. For example, the cholinergic and dopamine systems that have been featured in other researchers' consideration of pupil size modulation are missing here. Thus, because the authors do not rule out plausible alternative hypotheses, the logical structure of this manuscript can be criticized as committing the fallacy of aOirming the consequent.

As we have noted in the response to the reviewer’s first point, we did not motivate our use of pupil size as an index of effort clearly enough. For the current purpose, the neural correlates of pupil size are less relevant than the cognitive correlates (see previous point). We reiterate that the neuromodulatory underpinnings of the observed pupil size effects (which indeed possibly include effects of the cholinergic, dopaminergic and serotonergic systems), while interesting for the discussion on the neural origin of effects, are not crucial to our conclusion. We hope the new rationale (without focusing too much on the (irrelevant) exact neural underpinnings) convinces the reviewer and reader.

Our changes to the manuscript are shown in our reply to the previous comment.

The reviewer notes that other plausible alternative hypotheses could explain the currently reported results. However, we did not find a more parsimonuous explanation for our data than ‘Effort Drives Saccade Selection’. Effort explains why participants prefer saccading toward specific directions in (1) highly controlled and (2) more natural settings. Note that we also predicted this effect previously (Koevoet et al., 2023). Moreover, this account explains (3) why participants make less saccades under additional cognitive demand, and (4) why especially costly saccades are reduced under additional cognitive demand. We are very open to the reviewer presenting other possible interpretations of our data so these can be discussed to be put to test in future work.

(3) The authors cite particular publications in support of the claim that saccade selection is influenced by an assessment of effort. Given the extensive work by others on this general topic, the skeptic could regard the theoretical perspective of this manuscript as too impoverished. Their work may be enhanced by consideration of other work on this general topic, e.g, (i) Shenhav A, Botvinick MM, Cohen JD. (2013) The expected value of control: an integrative theory of anterior cingulate cortex function. Neuron. 2013 Jul 24;79(2):217-40. (ii) Müller T, Husain M, Apps MAJ. (2022) Preferences for seeking effort or reward information bias the willingness to work. Sci Rep. 2022 Nov 14;12(1):19486. (iii) Bustamante LA, Oshinowo T, Lee JR, Tong E, Burton AR, Shenhav A, Cohen JD, Daw ND. (2023) Effort Foraging Task reveals a positive correlation between individual differences in the cost of cognitive and physical effort in humans. Proc Natl Acad Sci U S A. 2023 Dec 12;120(50):e2221510120.

We thank the reviewer for pointing us toward this literature. These papers are indeed relevant for our manuscript, and we have now incorporated them. Specifically, we now discuss how the costs of effort are weighed in relation to possible rewards during decision-making. We have also incorporated work that has investigated how the biomechanical costs of arm movements contribute to action selection.

“Our findings are in line with established effort-based models that assume costs to be weighed against rewards during decision-making [102–107]. In such studies, reward and cognitive/physical effort are often parametrically manipulated to as- sess how much effort participants are willing to exert to acquire a given (monetary) reward [e.g. 108, 109]. Whereas this line of work manipulated the extrinsic costs and/or rewards of decision options (e.g. perceptual consequences of saccades [110, 111] or consequences associated with decision options), we here focus on the intrin- sic costs of the movement itself (in terms of cognitive and physical effort). Relatedly, the intrinsic costs of arm movements are also considered during decision-making: biomechanically aOordable movements are generally preferred over more costly ones [26–28]. We here extend these findings in two important ways. First, until now, the intrinsic costs of saccades and other movements have been inferred from gaze behavior itself or by using computational modelling [23, 25–28, 34, 35, 112]. In con- trast, we directly measured cost physiologically using pupil size. Secondly, we show that physiologically measured saccade costs predict where saccades are directed in a controlled binary preference task, and even during natural viewing. Our findings could unite state-of-the-art computational models [e.g. 23, 25, 34, 35, 113] with physiological data, to directly test the role of saccade costs and ultimately further our understanding of saccade selection.”

(4) What is the source of cost in saccade production? What is the currency of that cost? The authors state (page 13), "... oblique saccades require more complex oculomotor programs than horizontal eye movements because more neuronal populations in the superior colliculus (SC) and frontal eye fields (FEF) [76-79], and more muscles are necessary to plan and execute the saccade [76, 80, 81]." This statement raises questions and concerns. First, the basis of the claim that more neurons in FEF and SC are needed for oblique versus cardinal saccades is not established in any of the publications cited. Second, the authors may be referring to the fact that oblique saccades require coordination between pontine and midbrain circuits. This must be clarified. Second, the cost is unlikely to originate in extraocular muscle fatigue because the muscle fibers are so different from skeletal muscles, being fundamentally less fatigable. Third, if net muscle contraction is the cost, then why are upward saccades, which require the eyelid, not more expensive than downward? Thus, just how some saccades are more effortful than others is not clear.

Unfortunately, our current data do not allow for the specification of what the source is of differences in saccade production, nor what the currency is. We want to explicitly state that while pupil size is a sensitive measure of saccade costs, pupil size cannot directly inform what underlying mechanisms are causing differences in saccade costs across conditions (e.g. directions). Nevertheless, we do speculate about these issues because they are important to consider. We thank the reviewer for pointing out the shortcomings in our initial speculations.

Broadly, we agree with the reviewer that a neural source of differences in costs between different types of saccades is more likely than a purely muscular account (also see Koevoet et al., 2023). Furthermore, we think that the observed differences in saccade costs for oblique vs. cardinal and up vs. down could be due to different underlying mechanisms. While we caution against overinterpreting single directions, tentative evidence for this may also be drawn by the different time course of effects for up/down versus cardinal/oblique, Figure 1c.

Below we speculate about why some specific saccade directions may be more costly than others:

Why would oblique saccades be more costly than cardinal saccades? We thank the reviewer for pointing out that oblique saccades additionally require coordination between pontine and midbrain circuits (Curthoys et al., 1984; King & Fuchs, 1979; Sparks, 2002). This point warrants more revised discussion compared to our initial version. We have incorporated this as follows:

“The complexity of an oculomotor program is arguably shaped by its neural underpinnings. For example, oblique but not cardinal saccades require communication between pontine and midbrain circuits [73–75]. Such differences in neural complexity may underlie the additional costs of oblique compared with cardinal saccades. Besides saccade direction, other properties of the ensuing saccade such as its speed, distance, curvature, and accuracy may contribute to a saccade’s total cost [22, 33, 53, 76, 77] but this remains to be investigated directly.”

Why would downward saccades be more costly than upward saccades? As the reviewer points out: from a net muscular contraction account of cost, one would expect the opposite pattern due to the movement of the eyelid. Instead, we speculate that our findings may be associated with the well-established anisotropy in early visual cortex along the vertical meridian. Specifically, the upper vertical meridian is represented at substantially less detail than the lower vertical meridian (Himmelberg et al., 2023; Silva et al., 2018). Prior to a saccade, attention is deployed towards the intended saccadic endpoint (Deubel & Schneider, 1996; Kowler et al., 1995). Attention tunes neurons to preferentially process the attended location over non-attended locations. Due to the fact that the lower visual field is represented at higher detail than the upper visual field, attention may tune neuronal responses differently when preparing up- compared with downward saccades (Hanning et al., 2024; Himmelberg et al., 2023). Thus, it may be more costly to prepare down- compared with upward saccades. This proposition, however, does not account for the lower costs associated horizontal compared with up- and downward saccades as the horizontal meridian is represented at a higher acuity than the vertical merdian. This makes it unlikely that this explains the pattern of results completely. Again, at this point we can only speculate why costs differ, yet we demonstrate that these differences in cost are decisive for oculomotor behavior. We now explicitly state the speculative nature of these ideas that would all need to be tested directly.

We have updated our discussion of this issue as follows:

“The observed differences in saccade costs across directions could be linked to established anisotropies in perception [80–86], attention [87–92], saccade charac- teristics [87, 88, 92, 93], and (early) visual cortex [94–98] [also see 99]. For example, downward saccades are more costly than upward saccades, which mimics a similar asymmetry in early visual areas wherein the upper visual field is relatively under- represented [94–98]; similarly stronger presaccadic benefits are found for down- compared with upward saccades [87, 88]. Moreover, upward saccades are more pre- cise than downward saccades [93]. Future work should elucidate where saccade cost or the aforementioned anisotropies originate from and how they are related - something that pupil size alone cannot address.”

(5) The authors do not consider observations about variation in pupil size that seem to be incompatible with the preferred hypothesis. For example, at least two studies have described systematically larger pupil dilation associated with faster relative to accurate performance in manual and saccade tasks (e.g., Naber M, Murphy P. Pupillometric investigation into the speed-accuracy trade-off in a visuo-motor aiming task. Psychophysiology. 2020 Mar;57(3):e13499; Reppert TR, Heitz RP, Schall JD. Neural mechanisms for executive control of speed-accuracy trade-off. Cell Rep. 2023 Nov 28;42(11):113422). Is the fast relative to the accurate option necessarily more costly?

We thank the reviewer for this interesting point that we will answer in two ways. First, we discuss the main point: the link between pupil size, effort, and cost. Second, we discuss the findings described specifically in these two papers and how we interpret these from a pupillometric account.

First, one may generally ask whether (1) any effort results in pupil dilation, (2) whether any effort is costly, and (3) whether this means that pupil dilation always reflects effort and cost respectively. Indeed, it has been argued repeatedly, prominently, and independently (e.g., Bumke, 1911; Mathôt, 2018) that any change in effort (no matter the specific origin) is associated with an evoked pupil dilation. Effort, in turn, is consistently and widely experienced as aversive, both across tasks and cultures (David et al., 2024). Effort minimization may therefore be seen as an universal law of human cognition and behavior with effort as a to-be minimized cost (Shadmehr et al., 2019; Hull 1943, Tsai 1932). However, this does not imply that any pupil dilation necessarily reflects effort or that, as a consequence thereof, any pupil dilation is always signaling cost. For instance, the pupil dark response, the pupil far response and changes in baseline pupil size are not associated with effort. Baseline and task-evoked pupil dilation responses have to be interpreted differently (see below), moreover, the pupil also changes (and dilates) due to other factors (see Strauch et al., 2022; Mathôt, 2018, Bumke 1911, Loewenfeld, 1999 for reviews).

Second, as for Naber & Murphy (2020) & Reppert at al. (2023) specifically: Both Reppert et al. (2023) and Naber & Murphy (2020) indeed demonstrate a larger *baseline* pupil size when participants made faster, less accurate responses. However, baseline pupil size is not an index of effort per-se, but task-evoked pupil dilation responses are (as studied in the present manuscript) (Strauch et al., 2022). For work on differences between baseline pupil diameter and task-evoked pupil responses, and their respective links with exploration and exploitation please see Jepma & Nieuwenhuis (2011). Indeed, the link between effort and larger pupil size holds for task evoked responses, but not baseline pupil size per se (also see Koevoet et al., 2023).

Still, Naber (third author of the current paper) & Murphy (2020) also demonstrated larger task-evoked pupil dilation responses when participants were instructed to make faster, less accurate responses compared with making accurate and relatively slow responses. However, this difference in task-evoked response gains significance only after the onset of the movement itself, and peaks substantially later than response offset. Whilst pupil dilation may be sluggish, it isn’t extremely sluggish either. As feedback to the performance of the participant was displayed 1.25s after performing the movement and clicking (taking about 630ms), we deem it possible that this effect may in part result from appraising the feedback to the participant rather than the speed of the response itself (in fact, Naber and Murphy also discuss this option). In addition to not measuring saccades but mouse movements, it is therefore possible that the observed evoked pupil effects in Naber & Murphy (2020) are not purely linked to motor preparation and execution per se. Therefore, future work that aims to investigate the costs of movements should isolate the effects of feedback and other potential factors that may drive changes in pupil size. This will help clarify whether fast or more accurate movements could be linked to the underlying costs of the movements.

Relatedly, we do not find evidence that pupil size during saccade planning predicts the onset latency of the ensuing saccade (please refer to our second response to Reviewer 2 for a detailed discussion).

Together, we therefore do not see the results from Reppert et al. (2023) and Naber & Murphy (2020) to be at odds with our interpretation of evoked pupil size reflecting effort and cost in the context of planning saccades.

We think that these are considerations important to the reader, which is why we now added them to the discussion as follows:

“Throughout this paper, we have used cost in the limited context of saccades.

However, cost-based decision-making may be a more general property of the brain [31, 36, 114–116]. Every action, be it physical or cognitive, is associated with an in- trinsic cost, and pupil size is likely a general marker of this [44]. Note, however, that pupil dilation does not always reflect cost, as the pupil dilates in response to many sensory and cognitive factors which should be controlled for, or at least considered, when interpreting pupillometric data [e.g., see 39, 40, 42, 117].”

(6) The authors draw conclusions based on trends across participants, but they should be more transparent about variation that contradicts these trends. In Figures 3 and 4 we see many participants producing behavior unlike most others. Who are they? Why do they look so different? Is it just noise, or do different participants adopt different policies?

We disagree with the transparency point of the reviewer. Note that we deviated from the norm here by being *more* transparent than common: we added individual data points and relationships rather than showing pooled effects across participants with error bars alone (see Figures 2c, 3b,c, 4c,e,f).

Moreover, our effects are consistent and stable across participants and are highly significant. To illustrate, for the classification analysis based on cost (Figure 2E) 16/20 participants showed an effect. As for the natural viewing experiments (total > 250,000 fixations), we also find that a majority of participants show the observed effects: Experiment 1: 15/16 participants; Experiment 2: 16/25 participants; Experiment 2 – adjustment: 22/25 participants.

We fully agree that it’s interesting to understand where interindividual variation may originate from. We currently have too little data to allow robust analyses across individuals and zooming in on individual differences in cost maps, preference maps, or potential personalized strategies of saccade selection. That said, future work could study this further. We would recommend to hereby reduce the number of directions to gain more pupil size data per direction and therefore cleaner signals that may be more informative on the individual level. With such stronger signals, studying (differences in) links on an individual level may be feasible and would be interesting to consider – and will be a future direction in our own work too. Nonetheless, we again stress that the reported effects are robust and consistent across participants, and that interindividual differences are therefore not extensive. Moreover, our results from four experiments consistently support our conclusion that effort drives saccade selection.

**Recommendations for the authors:**

**Reviewer #1 (Recommendations For The Authors):**
- Based on the public review, I would recommend that the authors carefully review and correct the manuscript with regard to the causal conclusions. The study is largely correlational (i.e. the pupil was only observed, not manipulated) and therefore does not allow causal conclusions to be drawn about the relationship between pupil size and saccade selection. These causal conclusions become even more confusing when pupil size is equated with effort and saccade cost. As a consequence, an actual correlation between pupil size and saccade selection has led to the title that effort drives saccade selection. It would also be helpful for the reader to summarize in an additional section of the discussion what they consider to be a causal or correlational link based on their results.

We agree with the reviewer, and we have indeed included more explicitly which findings are correlational and which causal in detail now. As outlined before we do not see a more parimanious explanation for our findings than our title, but we fully agree that the paper benefits from making the correlational/causal nature of evidence for this idea explicitly transparent.

“We report a combination of correlational and causal findings. Despite the correlational nature of some of our results, they consistently support the hypothesis that saccade costs predicts saccade selection [which we predicted previously, 33]. Causal evidence was provided by the dual-task experiment as saccade frequencies - and especially costly saccades were reduced under additional cognitive demand. Only a cost account predicts (1) a link between pupil size and saccade preferences, (2) a cardinal saccade bias, (3) reduced saccade frequency under additional cognitive demand, and (4) disproportional cutting of especially those directions associated with more pupil dilation. Together, our findings converge upon the conclusion that effort drives saccade selection.”

- Can the authors please elaborate in more detail on how they transformed the predictors of their linear mixed model for the visualization in Figure 1f? It is difficult to see how the coeOicients in the table and the figure match.

We used the ‘effectsize’ package to provide effect sizes of for each predictor of the linear mixed-effects model (https://cran.r-project.org/web/packages/effectsize/index.html). We report absolute effect sizes to make it visually easier to compare different predictors. These details have now been included in the Methods section to be more transparent about how these effect sizes were computed.

“Absolute effect sizes (i.e. *r*) and their corresponding 95% confidence intervals for the linear mixed-effects models were calculated using *t* and *df* values with the ’effectsize’ package (v0.8.8) in R.”

- Could the authors please explain in more detail why they think that a trial-by-trial analysis in the free choice task adds something new to their conclusions? In fact, a trialby-trial analysis somehow suggests that the pupil size data would enter the analysis at a single trial level. If I understand correctly, the pupil size data come from their initial mapping task. So there is only one mean pupil size for a given participant and direction that goes into their analysis to predict free choice in a single trial. If this is the case, I don't see the point of doing this additional analysis given the results shown in Figure 2c.

The reviewer understands correctly that pupil size data is taken from the initial mapping task. We then used these mean values to predict which saccade target would be selected on a trial-by-trial basis. While showing the same conceptual result as the correlation analysis, we opted to include this analysis to show the robustness of the results across individuals. Therefore we have chosen to keep the analysis in the manuscript but now write more clearly that this shows the same conceptual finding as the correlation analysis.

“As another test of the robustness of the effect, we analyzed whether saccade costs predicted saccade selection on a trial-by-trial basis. To this end, we first determined the more aOordable option for each trial using the established saccade cost map (Figure 1d). We predicted that participants would select the more aOordable option. Complementing the above analyses, the more aOordable option was chosen above chance level across participants (M = 56.64%, 95%-CI = [52.75%-60.52%], one-sample t-test against 50%: t(19) = 3.26, p = .004, Cohen’s d = .729; Figure 2e). Together, these analyses established that saccade costs robustly predict saccade preferences.”

**Reviewer #2 (Recommendations For The Authors):**
The authors report that "Whenever the difference in pupil size between the two options was larger, saccades curved away more from the non-selected option (β = .004, SE = .001, t = 4.448, p < .001; Figure 3b), and their latencies slowed (β = .050, SE = .013, t = 4.323, p < .001; Figure 3c)". I suspect this effect might not be driven by the difference but by a correlation between pupil size and latency.The authors correlate differences in pupil size (Exp1) with saccade latencies (Exp2), I recommend correlating pupil size with the latency directly, in either task. This would show if it is actually the difference between choices or simply the pupil size of the respective individual option that is linked to latency/effort. Same for curvature.

The reviewer raises a good point. Please see the previous analyses concerning the possible correlations between pupil size and saccade latency, and how they jointly predict saccade selection.

Our data show that saccade curvature and latencies are linked with the difference in pupil size between the selected and non-selected options. Are these effects driven by a difference in pupil size or by the pupil size associated with the chosen option?

To assess this, we conducted two linear mixed-effects models. We predicted saccade curvature and latency using pupil size (from the planning task) of the selected and nonselected options while controlling for the chosen direction (Wilkinson notation: saccade curvature/latency ~ selected pupil size + non-selected pupil size + obliqueness + vertical + horizontal + (1+ selected pupil size + non-selected pupil size|participant)). We found that saccades curved away more from costlier the non-selected targets (β = 1.534, *t* = 8.151, *p* < .001), and saccades curved away from the non-selected target less when the selected target was cheaper (β = -2.571, *t* = -6.602, *p* < .001). As the costs of the selected and non-selected show opposite effects on saccade curvature, this indicates that the difference between the two options drives oculomotor conflict.

As for saccade latencies, we found saccade onsets to slow when the cost of the selected target was higher (b = .068, *t* = 2.844, *p* = .004). In contrast, saccade latencies were not significantly affected by the cost of the non-selected target (β = -.018, *t* = 1.457, *p* = .145), although numerically the effect was in the opposite direction. This shows that latencies were primarily driven by the cost of the selected target but a difference account cannot be fully ruled out.

Together, these analyses demonstrate that the difference in costs between two alternatives reliably affects oculomotor conflict as indicated by the curvature analysis. However, saccade latencies are predominantly affected by the cost of the selected target – even when controlling for the obliqueness, updownness and leftrightness of the ensuing saccade. We have added these analyses here for completeness, but because the findings seem inconclusive for saccade latency we have chosen to not include these analyses in the current paper. We are open to including these analyses in the supplementary materials if the reviewer and/or editor would like us to, but have chosen not to do so due to conciseness and to keep the paper focused.

I was wondering why the authors haven't analyzed the pupil size in Experiment 2. If the pupil size can be assessed during a free viewing task (Experiment 3), shouldn't it be possible to also evaluate it in the saccade choice task?

We did not analyze the pupil size data from the saccade preference task for two reasons. First, the number of saccades is much lower than in the natural search experiments (~14.000 vs. ~250.000). Second, in the saccade preference task, there were always two possible saccade targets. Therefore, even if we were able to isolate an effort signal, this signal could index a multitude of factors such as deciding between two possible saccade targets (de Gee et al., 2014), and has the possibility of two oculomotor programs being realized instead of only a single one (Van der Stigchel, 2010).

Discussion: "due to stronger presaccadic benefits for upward compared with downward saccades [93,94]". I think this should be the other way around.

We thank the reviewer for pointing this out. We have corrected our mistake in the revised manuscript.

Saccade latencies differ around the visual field; to account for that, results / pupil size should be (additionally) evaluated relative to saccade onset (rather than cue offset). It is interesting that latencies were not accounted for here (Exp1), since they are considered for Exp2 (where they correlate with a pupil size difference). I suspect that latencies not only correlate with the difference in pupil size, but directly with pupil size itself.

We agree with the reviewer that locking the pupil size signal to saccade onset instead of cue offset may be informative. We included an analysis in the supporting information that investigates this (see Figure S1). The results of the analysis were conceptually identical.

The reviewer writes that latencies were not accounted for in Experiment 1. Although saccade latency was not included in the final model reported in the paper, it was considered during AIC-based backward model selection. As saccade latency did not predict meaningful variance in pupil size, it was ultimately not included in the analysis as a predictor. For completeness, we here report the outcome of a linear mixed-effects that does include saccade latency as a predictor. Here, saccade latencies did not predict pupil size (β = 1.859e-03, *t* = .138, *p* = .889). The assymetry effects remained qualitatively unchanged: preparing oblique compared with cardinal saccades resulted in a larger pupil size (β = 7.635, *t* = 3.969, *p* < .001), and preparing downward compared with upward saccades also led to a larger pupil size (β = 3.344, *t* = 3.334, *p* = .003).

In addition, we have included a new analysis in the supporting information that directly addresses this issue. We will reiterate the main results here:

“To ascertain whether pupil size or other oculomotor metrics predict saccade preferences, we conducted a multiple regression analysis. We calculated average pupil size, saccade latency, landing precision and peak velocity maps across all 36 directions. The model, determined using AIC-based backward selection, included pupil size, latency and landing precision as predictors (Wilkinson notation: saccade preferences pupil size + saccade latency + landing precision). The analysis re- vealed that pupil size (β = -42.853, t = 4.791, p < .001) and saccade latency (β = -.377, t = 2.106, p = .043) predicted saccade preferences. Landing precision did not reach significance (β = 23.631, t = 1.675, p = .104). Together, this demonstrates that although other oculomotor metrics such as saccade latency contribute to saccade selection, pupil size remains a robust marker of saccade selection.”

We have also added this point in our discussion:

“We here measured cost as the degree of effort-linked pupil dilation. In addition to pupil size, other markers may also indicate saccade costs. For example, saccade latency has been proposed to index oculomotor effort [100], whereby saccades with longer latencies are associated with more oculomotor effort. This makes saccade latency a possible complementary marker of saccade costs (also see Supplemen- tary Materials). Although relatively sluggish, pupil size is a valuable measure of attentional costs for (at least) two reasons. First, pupil size is a highly established as marker of effort, and is sensitive to effort more broadly than only in the context of saccades [36–45, 48]. Pupil size therefore allows to capture not only the costs of saccades, but also of covert attentional shifts [33], or shifts with other effectors such as head or arm movements [54, 101]. Second, as we have demonstrated, pupil size can measure saccade costs even when searching in natural scenes (Figure 4). During natural viewing, it is difficult to disentangle fixation duration from saccade latencies, complicating the use of saccade latency as a measure of saccade cost. Together, pupil size, saccade latency, and potential other markers of saccade cost could fulfill complementary roles in studying the role of cost in saccade selection.”

References

Alnæs, D., Sneve, M. H., Espeseth, T., Endestad, T., van de Pavert, S. H. P., & Laeng, B. (2014). Pupil size signals mental eFort deployed during multiple object tracking and predicts brain activity in the dorsal attention network and the locus coeruleus. Journal of Vision, 14(4), 1. https://doi.org/10.1167/14.4.1

Awh, E., Belopolsky, A. V., & Theeuwes, J. (2012). Top-down versus bottom-up attentional control: A failed theoretical dichotomy. Trends in Cognitive Sciences, 16(8), 437–443. https://doi.org/10.1016/j.tics.2012.06.010

Ballard, D. H., Hayhoe, M. M., & Pelz, J. B. (1995). Memory Representations in Natural Tasks. Journal of Cognitive Neuroscience, 7(1), 66–80. https://doi.org/10.1162/jocn.1995.7.1.66

Beatty, J. (1982). Task-evoked pupillary responses, processing load, and the structure of processing resources. Psychological Bulletin, 91(2), 276–292. https://doi.org/10.1037/0033-2909.91.2.276

Bumke, O. (1911). Die Pupillenstörungen bei Geistes-und Nervenkrankheiten (2nd ed.). Fischer.

Curthoys, I. S., Markham, C. H., & Furuya, N. (1984). Direct projection of pause neurons to nystagmusrelated excitatory burst neurons in the cat pontine reticular formation. Experimental Neurology, 83(2), 414–422. https://doi.org/10.1016/S0014-4886(84)90109-2

David, L., Vassena, E., & Bijleveld, E. (2024). The unpleasantness of thinking: A meta-analytic review of the association between mental eFort and negative aFect. Psychological Bulletin, 150(9), 1070–1093. https://doi.org/10.1037/bul0000443

de Gee, J. W., Knapen, T., & Donner, T. H. (2014). Decision-related pupil dilation reflects upcoming choice and individual bias. Proceedings of the National Academy of Sciences, 111(5), E618–E625. https://doi.org/10.1073/pnas.1317557111

Deubel, H., & Schneider, W. X. (1996). Saccade target selection and object recognition: Evidence for a common attentional mechanism. Vision Research, 36(12), 1827–1837. https://doi.org/10.1016/0042-6989(95)00294-4

Greenwood, J. A., Szinte, M., Sayim, B., & Cavanagh, P. (2017). Variations in crowding, saccadic precision, and spatial localization reveal the shared topology of spatial vision. Proceedings of the National Academy of Sciences, 114(17), E3573–E3582. https://doi.org/10.1073/pnas.1615504114

Hanning, N. M., Himmelberg, M. M., & Carrasco, M. (2024). Presaccadic Attention Depends on Eye Movement Direction and Is Related to V1 Cortical Magnification. Journal of Neuroscience, 44(12). https://doi.org/10.1523/JNEUROSCI.1023-23.2023

Himmelberg, M. M., Winawer, J., & Carrasco, M. (2023). Polar angle asymmetries in visual perception and neural architecture. Trends in Neurosciences, 46(6), 445–458. https://doi.org/10.1016/j.tins.2023.03.006

Jepma, M., & Nieuwenhuis, S. (2011). Pupil Diameter Predicts Changes in the Exploration–Exploitation Trade-oF: Evidence for the Adaptive Gain Theory. Journal of Cognitive Neuroscience, 23(7), 1587– 1596. https://doi.org/10.1162/jocn.2010.21548

Kahneman, D. (1973). Attention and Effort. Prentice-Hall.

Kahneman, D., & Beatty, J. (1966). Pupil diameter and load on memory. Science (New York, N.Y.), 154(3756), 1583–1585. https://doi.org/10.1126/science.154.3756.1583

King, W. M., & Fuchs, A. F. (1979). Reticular control of vertical saccadic eye movements by mesencephalic burst neurons. Journal of Neurophysiology, 42(3), 861–876. https://doi.org/10.1152/jn.1979.42.3.861

Koevoet, D., Strauch, C., Naber, M., & Van der Stigchel, S. (2023). The Costs of Paying Overt and Covert Attention Assessed With Pupillometry. Psychological Science, 34(8), 887–898. https://doi.org/10.1177/09567976231179378

Koevoet, D., Strauch, C., Van der Stigchel, S., Mathôt, S., & Naber, M. (2024). Revealing visual working memory operations with pupillometry: Encoding, maintenance, and prioritization. WIREs Cognitive Science, e1668. https://doi.org/10.1002/wcs.1668

Kowler, E., Anderson, E., Dosher, B., & Blaser, E. (1995). The role of attention in the programming of saccades. Vision Research, 35(13), 1897–1916. https://doi.org/10.1016/0042-6989(94)00279-U

Laeng, B., Sirois, S., & Gredebäck, G. (2012). Pupillometry: A Window to the Preconscious? Perspectives on Psychological Science, 7(1), 18–27. https://doi.org/10.1177/1745691611427305

Loewenfeld, I. E. (1958). Mechanisms of reflex dilatation of the pupil. Documenta Ophthalmologica, 12(1), 185–448. https://doi.org/10.1007/BF00913471

Mathôt, S. (2018). Pupillometry: Psychology, Physiology, and Function. Journal of Cognition, 1(1), 16. https://doi.org/10.5334/joc.18

Naber, M., & Murphy, P. (2020). Pupillometric investigation into the speed-accuracy trade-oF in a visuomotor aiming task. Psychophysiology, 57(3), e13499. https://doi.org/10.1111/psyp.13499

Nozari, N., & Martin, R. C. (2024). Is working memory domain-general or domain-specific? Trends in Cognitive Sciences, 0(0). https://doi.org/10.1016/j.tics.2024.06.006

Reppert, T. R., Heitz, R. P., & Schall, J. D. (2023). Neural mechanisms for executive control of speedaccuracy trade-oF. Cell Reports, 42(11). https://doi.org/10.1016/j.celrep.2023.113422

Richer, F., & Beatty, J. (1985). Pupillary Dilations in Movement Preparation and Execution. Psychophysiology, 22(2), 204–207. https://doi.org/10.1111/j.1469-8986.1985.tb01587.x

Robison, M. K., & Brewer, G. A. (2020). Individual diFerences in working memory capacity and the regulation of arousal. Attention, Perception, & Psychophysics, 82(7), 3273–3290. https://doi.org/10.3758/s13414-020-02077-0

Robison, M. K., & Unsworth, N. (2019). Pupillometry tracks fluctuations in working memory performance. Attention, Perception, & Psychophysics, 81(2), 407–419. https://doi.org/10.3758/s13414-0181618-4

Sahakian, A., Gayet, S., PaFen, C. L. E., & Van der Stigchel, S. (2023). Mountains of memory in a sea of uncertainty: Sampling the external world despite useful information in visual working memory. Cognition, 234, 105381. https://doi.org/10.1016/j.cognition.2023.105381

Shadmehr, R., Reppert, T. R., Summerside, E. M., Yoon, T., & Ahmed, A. A. (2019). Movement Vigor as a Reflection of Subjective Economic Utility. Trends in Neurosciences, 42(5), 323–336. https://doi.org/10.1016/j.tins.2019.02.003

Silva, M. F., Brascamp, J. W., Ferreira, S., Castelo-Branco, M., Dumoulin, S. O., & Harvey, B. M. (2018). Radial asymmetries in population receptive field size and cortical magnification factor in early visual cortex. NeuroImage, 167, 41–52. https://doi.org/10.1016/j.neuroimage.2017.11.021

Sirois, S., & Brisson, J. (2014). Pupillometry. WIREs Cognitive Science, 5(6), 679–692. https://doi.org/10.1002/wcs.1323

Sparks, D. L. (2002). The brainstem control of saccadic eye movements. Nature Reviews Neuroscience, 3(12), Article 12. https://doi.org/10.1038/nrn986

Strauch, C., Wang, C.-A., Einhäuser, W., Van der Stigchel, S., & Naber, M. (2022). Pupillometry as an integrated readout of distinct attentional networks. Trends in Neurosciences, 45(8), 635–647. https://doi.org/10.1016/j.tins.2022.05.003

Unsworth, N., & Miller, A. L. (2021). Individual DiFerences in the Intensity and Consistency of Attention. Current Directions in Psychological Science, 30(5), 391–400. https://doi.org/10.1177/09637214211030266

Van der Stigchel, S. (2010). Recent advances in the study of saccade trajectory deviations. Vision Research, 50(17), 1619–1627. https://doi.org/10.1016/j.visres.2010.05.028

Van der Stigchel, S. (2020). An embodied account of visual working memory. Visual Cognition, 28(5–8), 414–419. https://doi.org/10.1080/13506285.2020.1742827

Van der Stigchel, S., & Hollingworth, A. (2018). Visuospatial Working Memory as a Fundamental Component of the Eye Movement System. Current Directions in Psychological Science, 27(2), 136–143. https://doi.org/10.1177/0963721417741710

van der Wel, P., & van Steenbergen, H. (2018). Pupil dilation as an index of eFort in cognitive control tasks: A review. Psychonomic Bulletin & Review, 25(6), 2005–2015. https://doi.org/10.3758/s13423-018-1432-y